# A Geometric Perspective towards Neural Calibration via Sensitivity Decomposition

**Junjiao Tian**
Georgia Institute of Technology
`jtian73@gatech.edu`

**Dylan Yung**
Georgia Institute of Technology
`dyung6@gatech.edu`

**Yen-Chang Hsu**
Samsung Research America
`yenchang.hsu@samsung.com`

**Zsolt Kira**
Georgia Institute of Technology
`zkira@gatech.edu`

## Abstract

It is well known that vision classification models suffer from poor calibration in the face of data distribution shifts. In this paper, we take a geometric approach to this problem. We propose *Geometric Sensitivity Decomposition (GSD)* which decomposes the norm of a sample feature embedding and the angular similarity to a target classifier into an *instance-dependent* and an *instance-independent* component. The instance-dependent component captures the sensitive information about changes in the input while the instance-independent component represents the insensitive information serving solely to minimize the loss on the training dataset. Inspired by the decomposition, we analytically derive a simple extension to current softmax-linear models, which learns to disentangle the two components during training. On several common vision models, the disentangled model outperforms other calibration methods on standard calibration metrics in the face of out-of-distribution (OOD) data and corruption with significantly less complexity. Specifically, we surpass the current state of the art by 30.8% relative improvement on corrupted CIFAR100 in Expected Calibration Error. Code available at `https://github.com/GT-RIPL/Geometric-Sensitivity-Decomposition.git`.

## 1 Introduction

During development, deep learning models are trained and validated on data from the same distribution. However, in the real world sensors degrade and weather conditions change. Similarly, subtle changes in image acquisition and processing can also lead to distribution shift of the input data. This is often known as *covariate shift*, and will typically decrease the performance (e.g. classification accuracy). However, it has been empirically found that the model's *confidence* remains high even when accuracy has degraded [1]. The process of aligning confidence to empirical accuracy is called model *calibration*. Calibrated probability provides valuable uncertainty information for decision making. For example, knowing when a decision cannot be trusted and more data is needed is important for safety and efficiency in real world applications such as self-driving [2] and active learning [3].

A comprehensive comparison of calibration methods has been studied for in-distribution (IND) data [4], However, these methods lead to unsatisfactory performance under distribution shift [5]. To resolve the problem, high-quality uncertainty estimation [6, 5] is required. Principled Bayesian methods [7] model uncertainty directly but are computationally heavy. Recent deterministic methods [8, 9] propose to improve a model's *sensitivity* to input changes by regularizing the model's intermediate layers. In this context, sensitivity is defined as preserving distance between two different input samples through layers of the model. We would like to utlize the improved sensitivity to better detect

Out-of-Distribution (OOD) data. However, these methods introduce added architecture changes and large combinatorics of hyperparameters.

Unlike existing works, we propose to study sensitivity from a geometric perspective. The last linear layer in a softmax-linear model can be decomposed into the multiplication of a norm and a cosine similarity term [10, 11, 12, 13]. Geometrically, the angular similarity dictates the membership of an input and the norm only affects the confidence in a softmax-linear model. Counter-intuitively, the norm of a sample's feature embedding exhibits little correlation to the hardness of the input [11]. Based on this observation, we explore two questions: 1) why is a model's confidence insensitive to distribution shift? 2) how do we improve model sensitivity and calibration?

We hypothesize that in part an insensitive norm is responsible for bad calibration especially on shifted data. We observe that the sensitivity of the angular similarity increases with training whereas the sensitivity of the norm remains low. More importantly, calibration worsens during the period when the norm increases while the angular similarity changes slowly. This shows a concrete example of the inability of the norm to *adapt* when accuracy has dropped. Intuitively, training on clean datasets encourages neural networks to *always* output increasingly large feature norm to continuously minimize the training loss. Because the probability of the prevalent class of an input is proportional to its norm, larger norms lead to smaller training loss when most training data have been classified correctly (See Sec. 3.1). This renders the norm insensitive to input differences because the model is trained to *always* output features with large norm on clean data. While we have put forth that the norm is poorly calibrated, we must emphasize that it can still play an important role in model calibration (See Sec. 4.1).

To encourage sensitivity, we propose to decompose the norm of a sample's feature embedding and the angular similarity into two components: *instance-dependent* and *instance-independent*. The instance-dependent component captures the sensitive information about the input while the instance-independent component represents the insensitive information serving solely to minimize the loss on the training dataset. Inspired by the decomposition, we analytically derive a simple extension to the current softmax-linear model, which learns to disentangle the two components during training. We show that our model outperforms other deterministic methods (despite their significant complexity) and is comparable to multi-pass methods with fewer training hyperparameters in Sec. 4.1.

In summary, our contributions are four fold:

- We study the problem of calibration geometrically and identify that the insensitive norm is responsible for bad calibration under distribution shift.
- We derive a principled but simple geometric decomposition that decomposes the norm into an instance-dependent and instance-independent component.
- Based on the decomposition, we propose a simple training and inference scheme to encourage the norm to reflect distribution changes.
- We achieve state of the art results in calibration metrics in the face of corruptions while having arguably the simplest calibration method to implement.

## 2 Related Work

Methods dedicated to strengthening calibration can be divided into two camps: multi-pass models and single-pass deterministic models. The current state-of-the-art multi-pass models are: Bayesian Monte Carlo Drop Out (MCDO) [7] and Deep Ensembles [14]. Bayesian methods are the most principled way to model uncertainty. Instead of *optimizing* max likelihood for a single set of parameters, Bayesian methods obtain a posterior distribution over possible parameters given a prior distribution over parameters and calculated data likelihood assuming some process noise. The posterior distribution over parameters captures epistemic uncertainty or uncertainty due to the limits of what the model knows . The final predictive distribution is obtained by *marginalizing* out model parameters. While Bayesian methods are theoretically sound, they are intractable in practice. Deep Ensembles averages multiple models trained using different random initialization so they learn different classification functions.

A recent trend is to use a single-pass deterministic **non-Bayesian** model to improve uncertainty estimation. Two recent works DUQ [8] and SNGP [9] propose to improve uncertainty-awarenesss of

deterministic networks by improving the networks' sensitivity to input changes. Intuitively, a sensitive model should map samples further from the training data as they become more out-of-distribution. This can be achieved at two levels: feature level and output level. At the feature level, both methods require the feature extractors (CNNs) to be regularized to prevent feature collapse, which is the mapping of two different data points to the same embedded vector. This is ensured by having input distance awareness, which is equivalent to ensuring bi-Lipschitz continuity [15].

In order to achieve this, DUQ [8] uses a two-sided gradient penalty [16] and SNGP [9] uses bounded spectral normalization [15]. The output level needs to reflect the changes in feature space. This can be done by adopting distance-aware classifiers. DUQ [8] uses a RBF networks with learned centroids for each class and SNGP [9] uses an approximate Gaussian Process layer. We were inspired by temperature scaling [4], which is another method for bettering calibration, but fails under distribution shift [5]. Our method does not require input distance awareness and instead leverages the geometric intuitions about the output layer, specifically properties of the norm of the input embedding.

## 3 Method

Following our hypothesise that the insensitivity of the norm is responsible for bad calibration on distribution shifted data, we propose geometric sensitivity decomposition (GSD) for the norm. We first introduce the geometric perspective of the last linear layer in Sec. 3.1 and then derive GSD in Sec. 3.2. To improve sensitivity of the norm and model calibration on shifted data, we propose a GSD-inspired training and inference procedure in Sec. 3.3 and Sec. 3.4.

### 3.1 Norm and Similarity

The output layer of a neural network can be written as a dot-product $< \mathbf{x}, \mathbf{w_y} >$, where $\mathbf{x}$ is the embedded input and $\mathbf{w_y}$ is the weight vector associated with class $y$. Though seemingly simple there are strong geometric and calibration related intuitions drawn from this. Several prior works [10, 12, 11] have studied the effects decomposition of the last linear layer in a softmax model can have on classification. The output layer can be decomposed into angular similarity $\cos \phi_y$ and norm $\|\mathbf{x}\|_2$.

$$P(y|x) = \frac{\exp l_y}{\sum_{j=1}^{c} \exp l_j} = \frac{\exp\left(\|\mathbf{w_y}\|_2 \|\mathbf{x}\|_2 \cos \phi_y\right)}{\sum_{j=1}^{c} \exp\left(\|\mathbf{w_j}\|_2 \|\mathbf{x}\|_2 \cos \phi_j\right)} \tag{1}$$

where $\|\mathbf{w_y}\|_2$ is the norm of a specific classifier in the linear layer. We'll use this geometric view of the linear layer instead of the dot-product representation.

Based on this perspective, we base the foundation of our work on the following observations from prior works [10, 12, 11]: 1) The probability/confidence of the prevalent class of an input is proportional to its norm [12]. 2) While the norm of a feature strongly scales the predictive probability, due to it's unregularized nature the norm is not sensitive to the hardness of the input [11]. In other words, the norm could be the reason for bad sensitivity of the confidence to input distribution shift. Consequently, the insensitive norm can be causally related to bad calibration. We will examine a strong correlation between the quality of calibration and the magnitude of norm in Sec. 4.2.

### 3.2 Geometric Sensitivity Decomposition of Norm and Angular Similarity

To motivate the subsequent geometric decomposition, we can revisit the softmax model, $P(y|x) \propto \exp\left(\|\mathbf{w_y}\|_2 \|\mathbf{x}\|_2 \cos \phi_y\right)$. There are three terms contributing to the magnitude of the exponential function, $\|\mathbf{w_y}\|_2$, $\|\mathbf{x}\|_2$ and $\cos \phi_y$. Due to weight regularizations, $\|\mathbf{w_y}\|_2$ is most likely very small, while $\cos \phi_y \in [-1, 1]$. Therefore, the only way to obtain a high probability/confidence on training data and minimize cross-entropy loss is to 1) push the norm $\|\mathbf{x}\|_2$ to a large value and 2) keep $\cos |\phi_y|$ of the ground truth class close to one, i.e., $|\phi_y|$ close to zero. This is further supported by [17], where it was shown that logits of the ground truth class must diverge to infinity in order to minimize cross-entropy loss under gradient descent. In this process, models tend towards *large* norms and *small* angles for all training samples.

Therefore, we propose to *decompose* the norms of features into two components: an *instance-independent* scalar offset and an *instance-dependent* variance factor, which we define in Eq. 2. The role of the instance-independent offset $\mathcal{C}_x$ is to minimize the loss on the entire training set and

the instance-dependent component $\Delta x$ accounts for differences in samples. Therefore, if we can disentangle the instance-independent component from the instance-dependent component, we can obtain a norm that is sensitive to the hardness of data. Following this logic, we decompose the norm into two components.

$$\|\mathbf{x}\|_2 = \|\Delta x\|_2 + \mathcal{C}_x \tag{2}$$

Similarly, we relax the angles such that the predicted angular similarity does not need to be close to one on the training data, i.e., making the angles larger. To achieve this, we introduce an instance-independent relaxation angle $\mathcal{C}_\phi$ and an instance-dependent angle $\Delta\phi_y$. Analogous to the norm decomposition, the scalar $\mathcal{C}_\phi$ serves solely to minimize the training loss while the instance-dependent $\Delta\phi_y$ accounts for differences in samples. Because we need to account for the sign of the angle, we put an absolute value on it.

$$|\phi_y| = |\Delta\phi_y| - |\mathcal{C}_\phi| \tag{3}$$

The $\|\Delta\mathbf{x}\|_2, |\Delta\phi_y|$ are the instance-dependent components and $\mathcal{C}_x, |\mathcal{C}_\phi|$ are the instance-independent components. We can rewrite the pre-softmax logits in Eq. 1 with the decomposed norm and angular similarity. (Detailed derivation in Sec. A.1 in the Appendix.)

$$\|\mathbf{x}\|_2 \cos\phi_y = \|\mathbf{x}\|_2 \cos|\phi_y| = (\|\Delta\mathbf{x}\|_2 + \mathcal{C}_x) \cos(|\Delta\phi_y| - |\mathcal{C}_\phi|) \tag{4}$$
$$= (\|\Delta\mathbf{x}\|_2 + \mathcal{C}_x) \frac{1}{\cos|\mathcal{C}_\phi|} \cos|\Delta\phi_y| \left(1 - \sin|\mathcal{C}_\phi|^2 \left(1 - \frac{\cos|\mathcal{C}_\phi|\sin|\Delta\phi_y|}{\sin|\mathcal{C}_\phi|\cos|\Delta\phi_y|}\right)\right)$$

We can simplify the equation by **assuming** $\cos|\phi_y|$ **is close to one, which means** $|\phi_y|$ **is small.** This is due to the fact that $|\phi_y|$ is the angle between the correct class weight and $x$, which means as training ensues, the angle converges to 0 and thus the cosine similarity converges to 1. (Please see Sec. A.2 for empirical support.)

$$\frac{\cos|\mathcal{C}_\phi|\sin|\Delta\phi_y|}{\sin|\mathcal{C}_\phi|\cos|\Delta\phi_y|} = \frac{\sin(|\Delta\phi_y| + |\mathcal{C}_\phi|) + \sin|\phi_y|}{\sin(|\Delta\phi_y| + |\mathcal{C}_\phi|) - \sin|\phi_y|} \approx 1 \tag{5}$$

Therefore, Eq. 4, omitting the absolute value on angles because *cos* is an even function, simplifies:

$$\|\mathbf{x}\|_2 \cos\phi_y \approx (\|\Delta\mathbf{x}\|_2 + \mathcal{C}_x) \frac{1}{\cos\mathcal{C}_\phi} \cos\Delta\phi_y \tag{6}$$
$$= \left(\frac{1}{\cos\mathcal{C}_\phi}\|\Delta\mathbf{x}\|_2 + \frac{1}{\cos\mathcal{C}_\phi}\mathcal{C}_x\right)\cos\Delta\phi_y$$
$$= \left(\frac{1}{\alpha}\|\Delta\mathbf{x}\|_2 + \frac{\beta}{\alpha}\right)\cos\Delta\phi_y$$

Because $\cos\mathcal{C}_\phi$ and $\mathcal{C}_x$ are instance-independent, we denote them as $\alpha$ and $\beta$ respectively. **This geometric decomposition of norm and cosine similarity inspires us to include $\alpha$ and $\beta$ as free trainable parameters in a new network and the network can learn to predict the more input-sensitive $\|\Delta\mathbf{x}\|_2$ and $\Delta\phi_y$ instead of the original $\|\mathbf{x}\|_2$ and $\phi_y$.** While both the angle and norm can be decomposed we direct the focus to the norm as the angle is *already* calibrated to accuracy [11]. In other words, angles have been shown to be sensitive to input changes in [11].

### 3.3 Disentangled Training

Following the derivation in Eq 6, we replace the norm, $\|\mathbf{x}\|_2$, in Eq. 1 by $\left(\frac{1}{\alpha}\|\Delta\mathbf{x}\|_2 + \frac{\beta}{\alpha}\right)$ and $\phi_y$ by $\Delta\phi_y$. $\|\Delta\mathbf{x}\|_2$ and $\Delta\phi_y$ are now learned outputs from a new network instead as shown in Eq. 6:

$$P(y|x) = \frac{\exp l_y}{\sum_{j=1}^c \exp l_j} = \frac{\exp\left(\|\mathbf{w_y}\|_2 \left(\frac{1}{\alpha}\|\Delta\mathbf{x}\|_2 + \frac{\beta}{\alpha}\right)\cos\Delta\phi_y\right)}{\sum_{j=1}^c \exp\left(\|\mathbf{w_j}\|_2 \left(\frac{1}{\alpha}\|\Delta\mathbf{x}\|_2 + \frac{\beta}{\alpha}\right)\cos\Delta\phi_j\right)} \tag{7}$$

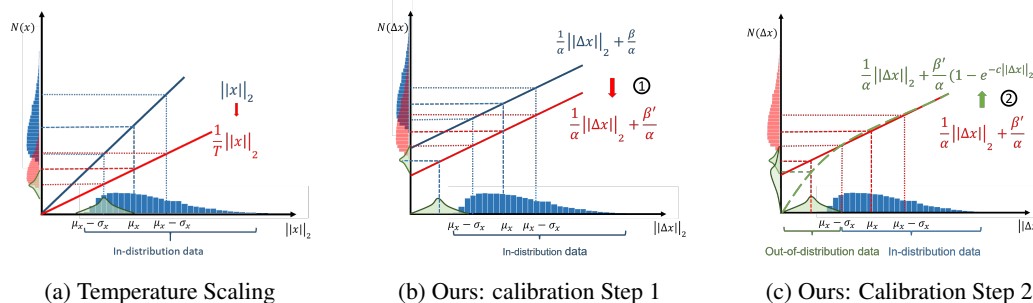

(a) Temperature Scaling  (b) Ours: calibration Step 1  (c) Ours: Calibration Step 2

Figure 1: **Calibration Procedure** (a): Temperature Scaling [4] changes the slope of the effective norm based on in-distribution (IND) data (See A.9 in Appendix)

The new model can be trained using the same training procedures as the vanilla network without additional hyperparameter tuning, changing the architecture or extended training time. Even though the outputs of the new network, $\|\Delta \mathbf{x}\|_2$ and $\Delta \phi_y$, only approximate the original geometric relationships with Eq. 6, the effect of $\alpha$ and $\beta$ reflects the decomposition in Eq. 3 and Eq. 2.

- $\beta$ encodes an instance-independent scalar $\mathcal{C}_x$ of the norm. A larger $\beta$ corresponds to a smaller instance-dependent component $\|\Delta \mathbf{x}\|_2$.
- $\alpha$ encodes the cosine of a relaxation angle $\mathcal{C}_\phi$. A larger $\arccos \alpha$ corresponds to a larger $\mathcal{C}_\phi$ and therefore a larger $\Delta \phi_j$.

Because $\beta$ encodes the independent component, the new feature norm $\|\Delta \mathbf{x}\|_2$ becomes sensitive to input changes and maps OOD data to lower norms than IND data as we can see in Fig. 3a, 3b. We regularize $\alpha$ such that the instance-independent component $\mathcal{C}_\phi$ is small. Specifically, we penalize $\|\alpha - 1\|_2^2$ because $\alpha = \cos \mathcal{C}_\phi$, i.e., if $\alpha \approx 1, \mathcal{C}_\phi \approx 0$. We empirically found that a larger relaxation angle $\mathcal{C}_\phi$ deteriorates performance because the angular similarity already correlates well with difficulty of data [11] and we do not need to encourage a large relaxation. Sec. 4.3 will empirically verify this.

### 3.4 Disentangled Inference

The decomposition theory in Sec. 3.2 provides a geometric perspective on the sensitivity of the norm and the angular similarity to input changes and inspires a disentangled model in Sec. 3.3. The new model uses a learnable affine transformation on the norm $\|\Delta \mathbf{x}\|_2$. Let's denote the affine transformed norm as the *effective norm* $\mathcal{N}(\Delta \mathbf{x}) \doteq \frac{1}{\alpha}\|\Delta \mathbf{x}\|_2 + \frac{\beta}{\alpha}$. However, the training only *separates* the sensitive components of the norm and angular similarity, the model can still be overconfident due to the existence of insensitive components. Therefore, we can improve calibration by modifying insensitive components, e.g., $\beta$ in our case. We propose a two-step calibration procedure that combines **in-distribution calibration** (Fig. 1b) and **out-of-distribution detection** (Fig. 1c) based on two observations: 1) overconfident IND data can be easily calibrated on a validation set, similar to temperature scaling [4]. 2) for OOD data, without access to a calibration set for OOD data, the best strategy is to map them far away from the IND data given that the model clearly distinguishes them.

**The first step** is calibrating the model on IND validation set (note our method does *not* rely on OOD validation data), similar to temperature calibration [4]. However, instead of tuning a temperature parameter as shown in Fig. 1a, we simply tune the offset parameter $\beta$ on the validation set in one of two ways: 1) grid-search based on minimizing Expected Calibration Error (see Sec. 4) 2) SGD optimization based on Negative Log Likelihood [4]. Because these are post-training procedure, both methods are very efficient. We denote the new parameter as $\beta'$. As shown in Fig. 1b, by changing the offset, we decrease the magnitude of the norms after the affine transformation. Formally,

$$\mathcal{N}(\Delta \mathbf{x}) = \frac{1}{\alpha}\|\Delta \mathbf{x}\|_2 + \frac{\beta}{\alpha} \rightarrow \mathcal{N}(\Delta \mathbf{x}) = \frac{1}{\alpha}\|\Delta \mathbf{x}\|_2 + \frac{\beta'}{\alpha} \tag{8}$$

**The second step** approximates the calibrated affine mapping in Eq. 8 by a non-linear function which covers a wider range of the effective norm as shown in Eq. 9 and maps OOD data further away from IND data. Intuitively, when a sample is more likely IND, the non-linear function maps it closer to the

calibrated transformation. When a sample is OOD, the non-linear function maps it more aggressively to a smaller magnitude, exponentially away from the IND samples.

$$\mathcal{N}(\Delta\mathbf{x}) = \frac{1}{\alpha}\|\Delta\mathbf{x}\|_2 + \frac{\beta'}{\alpha}(1 - e^{-c\|\Delta\mathbf{x}\|_2}) \tag{9}$$

where $c$ is a hyperparameter which can be calculated as in Eq. 10. The non-linear function grows exponentially close to the calibrated affine mapping in Eq. 8 dictated by $1 - e^{-c\|\Delta\mathbf{x}\|_2}$ as shown in 1c. Therefore, $e^{-c\|\Delta\mathbf{x}\|_2}$ can be viewed as an *error* term that quantifies how close the non-linear function is to the calibrated affine function in Eq. 8. Let $\mu_x$ and $\sigma_x$ denote the mean and standard deviation of the distribution of the norm of IND sample embedding calculated on the validation set. We use the heuristic that when evaluated at one standard deviation below the mean, $\|\Delta\mathbf{x}\|_2 = \mu_x - \sigma_x$, the approximation error $e^{-c(\mu_x - \sigma_x)} = 0.1$. Even though the error threshold is a hyperparameter, using an error of 0.1 lead to state-of-the-art results across all models applied.

$$c = \frac{-ln(1 - error)}{\mu_x - \sigma_x} = \frac{-ln(0.9)}{\mu_x - \sigma_x} \tag{10}$$

In summary, the sensitive norm $\|\Delta\mathbf{x}\|_2$ is used both as a soft threshold for OOD detection and as a criterion for calibration. While similar post-processing calibration procedure exists, such as temperature scaling [4] (illustrated in Fig. 1a and further introduced in A.9) it only provides good calibration on IND data and does not provide any mechanism to improve calibration on shifted data [5]. Our calibration procedure can improve calibration on both IND and OOD data, without access to OOD data, because the training method extracts the sensitive component in a principled manner. Just as temperature scaling, the non-linear mapping needs only to be calculated *once* and adds no computation at inference.

## 4 Experiments

### 4.1 Experiments on Calibration

Table 1: **ResNet-28-10 on CIFAR10** averaged over 10 seed. † denotes results from [9]. Our method outperforms other single-pass methods and is comparable to Deep Ensemble [14] on corrupted data. While the ensembled version of our model beats all multi-pass models.

| | Method | Accuracy ↑ | | ECE ↓ | | NLL ↓ | |
|---|---|---|---|---|---|---|---|
| | | Clean | Corrupted | Clean | Corrupted | Clean | Corrupted |
| Single-Pass | Vanilla† | **96.0±0.01** | 72.9±0.01 | 0.023±0.002 | 0.153±0.011 | 0.158±0.01 | 1.059±0.02 |
| | DUQ† | 94.7±0.02 | 71.6±0.02 | 0.034±0.002 | 0.183±0.011 | 0.239±0.02 | 1.348±0.01 |
| | SNGP† | 95.9±0.01 | 74.6±0.01 | 0.018±0.001 | 0.090±0.012 | **0.138±0.01** | 0.935±0.01 |
| | Ours $\beta'$ Grid-Searched | 95.9±0.01 | **74.9±0.05** | 0.018±0.003 | **0.067±0.010** | 0.148±0.003 | **0.826±0.03** |
| | Ours $\beta'$ Optimized | 95.9±0.01 | **74.9±0.05** | **0.008±0.002** | 0.085±0.012 | 0.140±0.004 | 0.853±0.04 |
| Multi-Pass | Deep Ensembles† | 96.6±0.01 | **77.9±0.01** | 0.010±0.001 | 0.087±0.004 | 0.114±0.01 | 0.815±0.01 |
| | MC Dropout† | 96.0±0.01 | 70.0±0.02 | 0.021±0.002 | 0.116±0.009 | 0.173±0.001 | 1.152±0.01 |
| | Ours $\beta'$Grid-Searched | **96.62** | **77.9** | **0.007** | **0.069** | **0.108** | **0.773** |

Table 2: **ResNet-28-10 on CIFAR100** averaged over 10 seeds. † denotes results from [9]. Our method outperforms other single-pass methods and Deep Ensemble [14] on corrupted data. While the ensembled version of our model beats all multi-pass models

| | Method† | Accuracy↑ | | ECE ↓ | | NLL ↓ | |
|---|---|---|---|---|---|---|---|
| | | Clean | Corrupted | Clean | Corrupted | Clean | Corrupted |
| Single-Pass | Vanilla† | 79.8±0.02 | **50.5±0.04** | 0.085±0.004 | 0.239±0.020 | 0.872±0.01 | 2.756±0.03 |
| | DUQ† | 78.5±0.02 | 50.4±0.02 | 0.119±0.001 | 0.281±0.012 | 0.980±0.02 | 2.841±0.01 |
| | SNGP† | **79.9±0.03** | 49.0±0.02 | **0.025±0.012** | 0.117±0.014 | 0.847±0.01 | 2.626±0.01 |
| | Ours $\beta'$ Grid-Searched | 79.8±0.03 | 49.8 ± 0.003 | 0.027±0.003 | **0.081 ± 0.007** | 0.787±0.009 | **2.23±0.02** |
| | Ours $\beta'$ Optimized | 79.8±0.03 | 49.8±0.03 | 0.027±0.003 | 0.088±0.007 | **0.784±0.011** | 2.236±0.021 |
| Multi-Pass | Deep Ensembles† | 80.2±0.01 | **54.1±0.04** | 0.021±0.004 | 0.138±0.013 | 0.666±0.02 | 2.281±0.03 |
| | MC Dropout† | 79.6±0.02 | 42.6±0.08 | 0.050±0.003 | 0.202±0.010 | 0.825±0.01 | 2.881±0.01 |
| | Ours $\beta'$ Grid-Searched | **83.09** | **54.1** | **0.018** | **0.086** | **0.614** | **2.042** |

The ultimate goal of the paper is to improve model calibration under distribution shift by improving sensitivity. Popular metrics for measuring calibration include: Negative Log-Likelihood (**NLL** [18]),

Table 3: **Generalizability Experiments** Our method is effective with different feature backbones.

| model | dataset | Clean | | | | Corrupt/Rotate | | | |
|---|---|---|---|---|---|---|---|---|---|
| | | accuracy↑ | ECE↓ | NLL↓ | Brier↓ | accuracy↑ | ECE↓ | NLL↓ | Brier↓ |
| ResNet34 | CIFAR10 | 95.63% | 0.026 | 0.186 | 0.007 | **81.96%** | 0.164 | 1.114 | 0.039 |
| GSD ResNet34 | CIFAR10 | **95.9%** | **0.005** | **0.148** | **0.006** | 76.54% | **0.088** | **0.882** | **0.037** |
| ResNet50 | CIFAR10 | 95.32% | 0.03 | 0.203 | 0.008 | **76.32%** | 0.17 | 1.23 | 0.039 |
| GSD ResNet50 | CIFAR10 | **95.82%** | **0.008** | **0.147** | **0.007** | 76.23% | **0.057** | **0.766** | **0.033** |
| ResNet101 | CIFAR10 | 95.61% | 0.028 | 0.197 | **0.007** | 77.59% | 0.154 | 1.118 | 0.037 |
| GSD ResNet101 | CIFAR10 | **95.62%** | **0.007** | **0.158** | **0.007** | **77.21%** | **0.075** | **0.852** | **0.036** |
| ResNet152 | CIFAR10 | **95.7%** | 0.028 | 0.196 | **0.007** | 75.2% | 0.179 | 1.337 | 0.041 |
| GSD ResNet152 | CIFAR10 | 95.63% | **0.007** | **0.151** | **0.007** | **76.58%** | **0.058** | **0.765** | **0.033** |
| ResNet34 | CIFAR100 | **78.81%** | 0.071 | **0.868** | 0.003 | **51.16%** | 0.19 | 2.387 | **0.007** |
| GSD ResNet34 | CIFAR100 | 78.02% | **0.037** | 0.938 | 0.003 | 49.27% | **0.098** | **2.361** | **0.007** |
| ResNet50 | CIFAR100 | **79.28%** | 0.075 | **0.861** | 0.003 | 49.71% | 0.213 | 2.477 | 0.007 |
| GSD ResNet50 | CIFAR100 | 78.97% | **0.033** | 0.879 | 0.003 | **50.12%** | **0.08** | **2.264** | **0.006** |
| ResNet101 | CIFAR100 | **80.17%** | 0.092 | 0.846 | 0.003 | **58.19%** | 0.253 | 2.575 | 0.007 |
| GSD ResNet101 | CIFAR100 | 79.82% | **0.034** | **0.834** | 0.003 | 53.14% | **0.082** | **2.11** | **0.006** |
| ResNet152 | CIFAR100 | **80.71%** | 0.090 | **0.815** | 0.003 | **54.2%** | 0.233 | 2.45 | 0.007 |
| GSD ResNet152 | CIFAR100 | 79.85% | **0.036** | 0.827 | 0.003 | 53% | **0.078** | **2.12** | **0.006** |

Table 4: **Importance of Norm** While norm is poorly calibrated, it is important for calibration.

| | ECE | NLL | Brier | Entropy | Accuracy |
|---|---|---|---|---|---|
| Vanilla ($\|\mathbf{w}_y\|\|\mathbf{x}\|\cos\phi_y$) | 0.025±0.001 | 0.186±0.006 | 0.001±0.0 | 0.082±0.002 | 95.4±0.1% |
| No Weight Norm (w/o $\|\mathbf{w}_y\|$) | 0.061±0.0003 | 0.206±0.006 | 0.001±0.0 | 0.527±0.014 | 95.4±0.1% |
| No $x$ Norm (w/o $\|\mathbf{x}\|$) | 0.893±0.002 | 2.837±0.005 | 0.009±0.0 | 4.537±0.001 | 95.4±0.1% |
| Only Cosine (w/o $\|\mathbf{w}_y\|,\|\mathbf{x}\|$) | 0.914±0.001 | 3.235±0.001 | 0.009±0.0 | 4.546±0.000 | 95.3±0.1% |

**Brier** [19] and Expected Calibration Error (**ECE** [20]). Our goal is for our model is to produce values close to 0 in these metrics, which maximizes calibration. Please refer to Sec. A.3 (Appendix) for more detailed discussion on these metrics. Following prior works [9, 8, 5], we will use CIFAR10 and CIFAR100 as the in-distribution training and testing dataset, and apply the image corruption library provided by [1] to benchmark calibration performance under distribution shift. The library provides 16 types of noises with 5 severity scales. In this section, we show that our model outperforms other deterministic methods (despite their significant complexity

**Compared Methods** We compare to several popular state-of-the-art models including stochastic Bayesian methods (multi-pass): Deep Ensemble [14] and MC dropout [7], and recent deterministic methods (single pass): SNGP [9] and DUQ [8].

**Results** In Tab. 1 and 2, we compare our model to the most recent state of art deterministic methods SNGP and DUQ using Wide ResNet 28-10 [21] as the model backbone and each model evaluated using the average of 10 seeds. We report accuracy, ECE and NLL on clean and corrupted CIFAR10/100 datasets [1]. Our method outperforms all single-pass methods on calibration when data is corrupted, and even surpass ensembles on error metrics for corrupted data. We had 2 versions of our model: **Grid Searched:** grid search $\beta'$ on the validation set to minimize ECE and **Optimized:** optimize $\beta'$ on the validation set via gradient decent to minimize NLL for 10 epochs, similar to temperature scaling. We report additional results with ResNet18 in Sec. A.4 and Sec. A.5 (Appendix) with image noise and rotation respectively.

**Generalizability** We explored how generalizable our method (Grid Searched) is by applying it to 12 different models and 4 different datasets in Tab. 3. We can see consistently that our model had stronger calibration across all models and metrics, including models known to be well calibrated like LeNet [22]. All models were tested on CIFAR10C and CIFAR100C datasets offered by [1] where the original CIFAR10 and CIFAR100 were pre-corrupted; these were used for consistent corruption benchmarking across all models. All non-CIFAR datasets were corrupted via rotation from angles [0,350] with 10 step angles in between and the average calibration and accuracy was taken across all

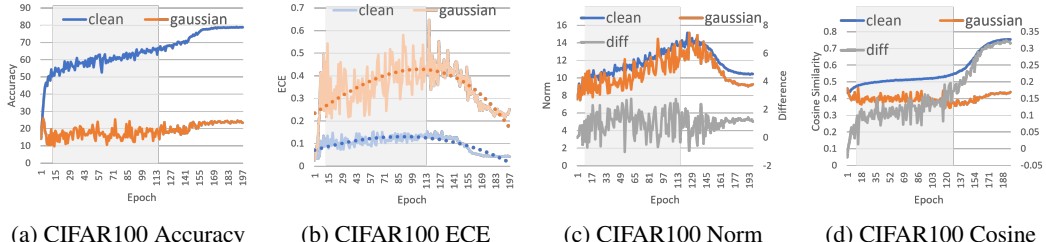

| (a) CIFAR100 Accuracy | (b) CIFAR100 ECE | (c) CIFAR100 Norm | (d) CIFAR100 Cosine |

Figure 2: **Accuracy, ECE, norm and cosine similarity on CIFAR100 validation set with clean and Gaussian noise trained on vanilla ResNet.** In the shaded region, increase in norm is responsible for increase in ECE because cosine similarity is relatively flat. Throughout training, sensitivity of the cosine similarity improves while that of the norm remains insensitive.

Table 5: **Pearson Correlation of Cosine Similarity and Norm vs. ECE during training on CIFAR100**. Norm is consistently positively correlated with ECE whereas the similarity is either negatively or not correlated with ECE.

|  | ResNet18 | | | ResNet34 | | | ResNet101 | | | ResNet152 | | |
|---|---|---|---|---|---|---|---|---|---|---|---|---|
|  | shot | Gaussian | Defocus | shot | Gaussian | Defocus | shot | Gaussian | Defocus | shot | Gaussian | Defocus |
| Cosine Sim | 0.09 | 0.03 | 0.73 | 0.09 | 0.03 | 0.32 | -0.03 | -0.04 | -0.88 | -0.97 | 0.04 | -0.81 |
| Norm | **0.82** | **0.82** | **0.78** | **0.82** | **0.81** | **0.78** | **0.87** | **0.87** | **0.85** | **0.86** | **0.85** | **0.81** |

degrees of rotation. Our models included: DenseNet [23], LeNet [22] and 6 varying sizes of ResNet, which are described in [24]. The datasets we experimented on CIFAR10 [25], CIFAR100 [25], MNIST [26] and SVHN [27], CIFAR10C [1], CIFAR100C [1]. We report Optimized results in Tab. 14 in A.7 (Appendix). Both tuning methods yield similar performance.

**Qualitative Comparison** The current state-of-the-art single pass models for inference on OOD data, without training on OOD data, are SNGP [9] and DUQ [8]. The primary disadvantages of these models are: **1) Hyperparameter Combinatorics:** Both DUQ and SNGP require many hyperparameters as shown in Tab. 13 in A.6 (Appendix). Our model only has *one* hyperparameter that is tuned post-training, which is quicker and less costly than the other methods that require pre-training tuning. **2) Extended Training Time:** DUQ requires a centroid embedding update every epoch, while SNGP requires sampling potentially high dimensional embeddings of training points, thus increasing training time while our model trains in the same amount of time as the model it is applied to. Bayesian MCDO [7] and Deep Ensemble [14] are considered the current state-of-the-art methods for multi-pass calibration. Bayesian MCDO requires multiple passes with dropout during inference. Deep Ensembles requires $N$ times the number of parameters as the single model it is ensembling where $N$ is the number of models ensembled. The main disadvantage of multi-pass models is high inference complexity while our model adds no overhead computation at inference.

**Importance of the Norm** While we have shown and conjectured that the norm of $x$ is uncalibrated to OOD data and not always well calibrated to IND data, one might suggest to simply remove the norm. We show in Tab. 4 though the norm is uncalibrated it is still important for inference. We trained ResNet18 on CIFAR10 and then ran inference with ResNet18 modified in the following: dividing out the norms of the weights for each class, dividing out the norm of the input and then dividing out both. As we can see the weight norm contributes minimally to inference as accuracy decreased by 0.03% without it and as previous work has shown the angle dominates classification. We can see with $||\mathbf{x}||$ removed the entropy is at it's highest while calibration is very poor, implying the distribution is much more uniform when it should be peaked, as a larger entropy implies a more uniform distribution. Thus the root of the issue does not lie in the existence of the norm, but it's lack of sensitivity.

## 4.2 Reasons for Bad Calibration under Distribution Shift

To identify the cause of bad calibration, we record the accuracy, ECE, norm and cosine similarity of a model during training of a vanilla ResNet model. Specifically, we record the evaluation statistics on clean data and also on data corrupted with Gaussian noise on CIFAR100. Fig. 2a and 2b show the accuracy and ECE respectively. We observe that evaluation on Gaussian noise corrupted data yields lower accuracy and higher ECE compared to evaluation on clean data. *This demonstrates that*

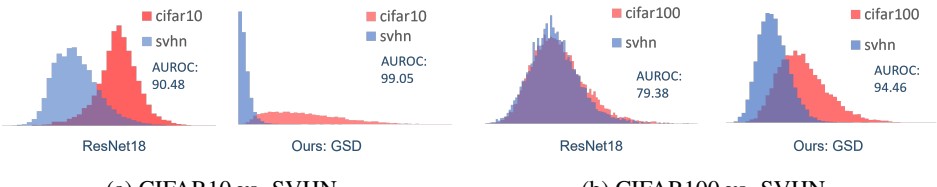

(a) CIFAR10 vs. SVHN           (b) CIFAR100 vs. SVHN

Figure 3: **Histogram of Norm Distribution** Our model ($\alpha$-regularized) improves separation of norm between IND and OOD data.

*the model's confidence fails to adapt to the decreasing accuracy.* Fig. 2c and 2d show the change of average norm and average cosine similarity throughout training. The difference between Gaussian noised data and clean data is also reported. We observe that the norm of clean data and the norm of Gaussian noised data are close and the difference remains constantly low whereas the cosine similarity of the two diverges with training. *This indicates that sensitivity of cosine similarity increases whereas sensitivity of the norm remains low with training.* In the shaded region of Fig. 2b-2d where ECE increases the most, we observe that the norm also increases but the cosine similarity only increases slowly. The observation also holds for other noises and architectures. We further present Pearson correlation between ECE and cosine similarity or norm on 4 models and 3 noises in Tab. 5. A large correlation coefficient indicates a higher positive correlation. Norm is consistently positively correlated with ECE whereas the similarity is either negatively or not correlated with ECE. This shows that the worsening of ECE (large ECE) is correlated with the increasing norm. Based on supporting literature [12], [11] and this correlation, the observation supports the conjecture that the insensitivity of the norm is responsible for bad calibration.

### 4.3 Empirical Support for the Disentangled Training

Table 6: **OOD AUROC↑ using Norm and Similarity** We show OOD detection results using norm and cosine similarity. SVHN [27] is used as the OOD dataset. Our method ($\alpha$-regularized) significantly increases the sensitivity of feature norm.

| ResNet18 | Criterion | CIFAR10 | CIFAR10 (Incorrect) |
|---|---|---|---|
| Vanilla | Norm | 90.48 | 67.23 |
|  | Similarity | 93.87 | 56.98 |
| $\alpha$- regularized | Norm | **99.05** | **93.16** |
|  | Similarity | 97.09 | 74.82 |
| $\alpha$- unregularized | Norm | 98.20 | 88.29 |
|  | Similarity | 94.72 | 60.63 |

| ResNet18 | Criterion | CIFAR100 | CIFAR100 (Incorrect) |
|---|---|---|---|
| vanilla | Norm | 79.38 | 62.66 |
|  | Similarity | 82.26 | 55.54 |
| $\alpha$- regularized | Norm | **94.46** | **86.67** |
|  | Similarity | 85.68 | 63.24 |
| $\alpha$- unregularized | Norm | 84.78 | 73.11 |
|  | Similarity | 72.61 | 42.90 |

(a) **CIFAR10 vs. SVHN AUROC**      (b) **CIFAR100 vs. SVHN AUROC**

In the first set of experiments, we show that $\alpha$ and $\beta$ reflect the effects of the geometric decomposition as claimed in Sec. 3.2 with different $\alpha - \beta$ configurations. From Fig. 4a - 4d, we observe that the norm decreases linearly with $\beta$ for fixed $\alpha$. From Fig. 4e - 4h, we observe that the angle increases linearly with $arccos(\alpha)$. The observations are consistent with the original geometric motivation. $\beta$ encodes an instance-independent portion, $\mathcal{C}_x$, of the norm. As $\beta$ increases, $\mathcal{C}_x$ increases and therefore the magnitude of the dependent component, $\|\Delta x\|_2$ decreases linearly. $\alpha$ encodes the inverse of the cosine of a relaxation angle, $\mathcal{C}_\phi$. As $arccos(\alpha)$ increases, the resulting angle, $\Delta\phi$ increases linearly due to the increased relaxation angle encoded by $\alpha$.

In the second set of experiments, we show that the new model effectively increases the sensitivity of both the norm and the angle to input distribution shift as claimed in Sec. 3.3. Specifically, we measure OOD detection performance of the models using both the norm and the cosine similarity with the Area Under the Receiver Operating Characteristic (AUROC) curve metric. We use CIFAR10/100 as

Table 7: **Average norm and accuracy across different corruptions on GSD ResNet18**. The table is organized in decreasing accuracy order.

| ResNet GSD | clean | brightness | fog | elastic | snow | defocus | frost | motion blur | jpeg | zoom blur | pixelate | contrast | shot | glass blur | impulse | Gaussian |
|---|---|---|---|---|---|---|---|---|---|---|---|---|---|---|---|---|
| accuracy | 95.33 | 93.82 | 88.75 | 85.09 | 83.87 | 82.95 | 80.41 | 79.71 | 79.31 | 78.48 | 76.4 | 75.29 | 59.46 | 59.29 | 57.26 | 47.33 |
| norm | 0.73 | 0.66 | 0.52 | 0.42 | 0.46 | 0.46 | 0.44 | 0.37 | 0.39 | 0.35 | 0.5 | 0.39 | 0.34 | 0.27 | 0.3 | 0.28 |

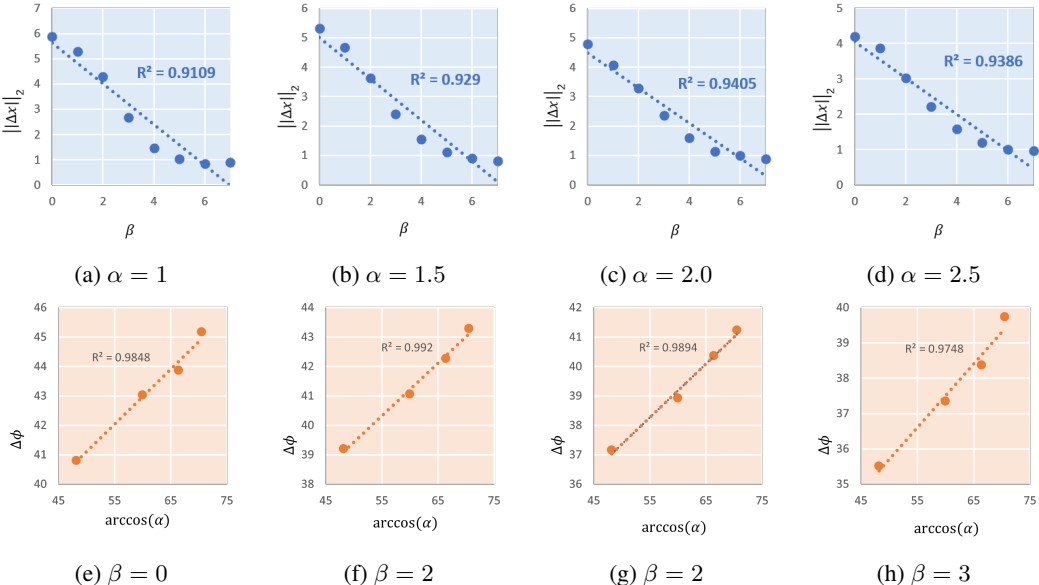

Figure 4: **Properties of** $\|\Delta x\|_2$ **and** $\Delta\phi$. (a) - (b): $\|\Delta x\|_2$ decreases linearly with $\beta$ for fixed $\alpha$ reflecting Eq. 2 and 6. (e) - (h) $\Delta\phi$ increases linearly with $arccos(\alpha)$ for fixed $\beta$ reflecting Eq. 3 and 6. All plots include R-squared values to indicate goodness-of-fit of the linear relationship.

the IND data and SVHN [27] as the OOD data. In Tab. 6a and 6b we show two configurations of models in addition to vanilla ResNet18: ($\alpha$-regularized) we regularize $\alpha$ such that it stays close to one as described in Sec. 3.3; ($\alpha$-unregularzed) we optimize both $\alpha$ and $\beta$ freely without constraints. Compared to vanilla ResNet, the norms predicted by our models achieve significant improvement in separating IND data from OOD data. Additionally, we visualize the distribution of norms in Fig. 3a and 3b. The separation between IND and OOD data increases significantly compared to vanilla ResNet18. However, a large $\alpha$ (see $\alpha$-unregularzed in Tab. 6a and 6b) leads to marginal cosine similarity sensitivity improvement on CIFAR10 and CIFAR100. This indirectly confirms our observations in Sec. 4.2 and in prior works [11] that cosine similarity correlates well with distribution shift. Introducing further angle relaxation might not be always beneficial. While we mainly focus on calibration, our method also strengthens its base model's ability for OOD detection.

The assumption that OOD data have smaller norms is based on the expectation that a model should be less confident on OOD data. Practically, the norm acts as a temperature in softmax as shown in Eq. 1. Intuitively, larger always yields more peaked/confident predictions, and smaller always yields flatter predictive distributions. Therefore, we expect less confident data such as OOD data to have smaller because we expect the output distribution to be flatter. The assumption is supported by the following empirical evidence. In Tab. 7 we show the norm of in-distribution and out-of-distribution data on CIFAR10 using ResNet50-GSD (ours). The OOD data is produced by the 15 corruptions used in the paper. OOD data have consistently smaller norms and the accuracy decreases with decreasing norm with a Pearson correlation of 0.9 as an indicator of more out-of-distribution.

## 5 Conclusion

In this paper, we studied the geometry of the last linear decision layer and identified the insensitivity of the norm as the culprit of bad calibration under distribution shift. To encourage sensitivity, we derived a general theory to decompose the norm and angular similarity. Inspired by the theory, we proposed a simple yet very effective training and inference scheme that encourages the norm to reflect distribution changes. The model outperforms other deterministic single pass-methods in calibration metrics with much fewer hyperparameters. We also demonstrated its superior generalizability on a variety of popular neural networks. Note that our problem and method have positive societal impact, as calibration under shift improves overall confidence and robustness of these models.

# 6 Acknowledgements

This work was partially supported by ONR grant N00014-18-1-2829.

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
