# A  Appendix

## A.1  Extended Derivation for Equation 4

In the main paper, we proposed to decompose the norm and angular similarity into instance-independent and dependent components.

$$\|\mathbf{x}\|_2 = \|\Delta x\|_2 + \mathcal{C}_x$$
$$|\phi_y| = |\Delta\phi_y| - |\mathcal{C}_\phi|$$

The $\|\Delta\mathbf{x}\|_2, |\Delta\phi_y|$ are the instance-dependent components and $\mathcal{C}_x, |\mathcal{C}_\phi|$ are the instance-independent components. We can rewrite the pre-softmax logits in Eq. 1 with the decomposed norm and angular similarity.

$$
\begin{aligned}
\|\mathbf{x}\|_2 \cos\phi_y = \|\mathbf{x}\|_2 \cos|\phi_y| &= (\|\Delta\mathbf{x}\|_2 + \mathcal{C}_x)\cos(|\Delta\phi_y| - |\mathcal{C}_\phi|) \\
&= (\|\Delta\mathbf{x}\|_2 + \mathcal{C}_x)(\cos|\Delta\phi_y|\cos|\mathcal{C}_\phi| + \sin|\Delta\phi_y|\sin|\mathcal{C}_\phi|) \\
&= (\|\Delta\mathbf{x}\|_2 + \mathcal{C}_x)\frac{1}{\cos|\mathcal{C}_\phi|}\left(\cos|\Delta\phi_y|\cos|\mathcal{C}_\phi|^2 + \sin|\Delta\phi_y|\cos|\mathcal{C}_\phi|\sin|\mathcal{C}_\phi|\right) \\
&= (\|\Delta\mathbf{x}\|_2 + \mathcal{C}_x)\frac{1}{\cos|\mathcal{C}_\phi|}\cos|\Delta\phi_y|\left(\cos|\mathcal{C}_\phi|^2 + \cos|\mathcal{C}_\phi|\sin|\mathcal{C}_\phi|\frac{\sin|\Delta\phi_y|}{\cos|\Delta\phi_y|}\right) \\
&= (\|\Delta\mathbf{x}\|_2 + \mathcal{C}_x)\frac{1}{\cos|\mathcal{C}_\phi|}\cos|\Delta\phi_y|\left(\left(1 - \sin|\mathcal{C}_\phi|^2\right) + \cos|\mathcal{C}_\phi|\sin|\mathcal{C}_\phi|\frac{\sin|\Delta\phi_y|}{\cos|\Delta\phi_y|}\right) \\
&= (\|\Delta\mathbf{x}\|_2 + \mathcal{C}_x)\frac{1}{\cos|\mathcal{C}_\phi|}\cos|\Delta\phi_y|\left(1 - \sin|\mathcal{C}_\phi|^2\left(1 - \underbrace{\frac{\cos|\mathcal{C}_\phi|\sin|\Delta\phi_y|}{\sin|\mathcal{C}_\phi|\cos|\Delta\phi_y|}}_{\approx 1 \text{ Eq. } 11}\right)\right) \\
&\approx \left(\frac{1}{\cos|\mathcal{C}_\phi|}\|\Delta\mathbf{x}\|_2 + \frac{\mathcal{C}_x}{\cos|\mathcal{C}_\phi|}\right)\cos|\Delta\phi_y|
\end{aligned}
$$

We can simplify the equation by **assuming $\cos|\phi_y|$ is close to one, which means $|\phi_y|$ is small.** This is due to the fact that $|\phi_y|$ is the angle between the correct class weight and $x$, which means as training ensues, the angle converges to 0 and thus the cosine similarity converges to 1. (Please see Sec. A.2 for empirical support.)

$$\frac{\cos|\mathcal{C}_\phi|\sin|\Delta\phi_y|}{\sin|\mathcal{C}_\phi|\cos|\Delta\phi_y|} = \frac{\sin(|\Delta\phi_y| + |\mathcal{C}_\phi|) + \sin|\phi_y|}{\sin(|\Delta\phi_y| + |\mathcal{C}_\phi|) - \sin|\phi_y|} \approx 1 \tag{11}$$

## A.2  Small Angle Assumption in Equation 5

Table 8: Average cosine similarity to the ground truth class on the training data set after training for 200 epochs

|  | CIFAR10 | | | CIFAR100 | | |
|---|---|---|---|---|---|---|
|  | ResNet-18 | ResNet-34 | ResNet-101 | ResNet-18 | ResNet-34 | ResNet-101 |
| $\cos\phi$ | 0.81 | 0.79 | 0.76 | 0.75 | 0.78 | 0.74 |

One reason for the small angle assumption in Eq. 5 is the observation that high-capacity models tend to be more miscalibrated [4] and our method is especially more effective in this case. When a model is sufficiently high-capacity compared to the diversity of the dataset, the assumption of small-angle is empirically more valid and the method can provide more significant improvement. All ResNet models are high-capacity deep models and corresponding cosine similarity to the true class is close to one during training as assumed in Sec. 3.2. Tab. 8 shows the average cosine similarity to the ground truth class on the training data.

### A.3 Definitions of Metrics

The problem tackled in this paper is supervised image classification in the face of noise. Assume a data point $X_i \in \mathbf{X}, i \in [1, N]$ each associated with a label $Y \in \mathbf{Y} = \{1, ..., K\}$. We would like our model $M$ where $M(X_i) = (\hat{Y}_i, \hat{P}_i)$ where $\hat{Y}_i$ is the class prediction and $\hat{P}$ is the probability/confidence given by the model to be as close to the ground truth distribution $P(Y_i|X_i)$. Ideally $\hat{P}_i$ is well calibrated which means that it represents the likelihood of the true event $\hat{Y}_i = Y_i$. *Perfect calibration* [4] can be defined as:

$$P(\hat{Y}_i = Y_i | \hat{P}_i = P_i) = P_i, \forall P_i \in [0, 1] \tag{12}$$

Ways of evaluating Calibration are as follows:

#### A.3.1 Expected Calibration Error (ECE)

Expected Calibration Error [20] evaluates calibration by calculating the difference in expectation between the confidence and accuracy or:

$$E_{\hat{P}}[|P(\hat{Y} = Y | \hat{P} = p) - p|] \tag{13}$$

This can also be computed as the weighted average of bins' accuracy/confidence difference:

$$\mathbf{ECE} = \sum_{m=1}^{M} \frac{|B_m|}{n} |accuracy(B_m) - confidence(B_m)| \tag{14}$$

where $n$ is the total number of samples. Perfect calibration is achieved bins when confidence equals accuracy and ECE $= 0$.

#### A.3.2 Negative Log Likelihood (NLL)

A way to measure a model's probabilistic quality is to use Negative Log Likelihood [18]. Given a probabilitist model $P(Y|X)$ and $N$ samples it is defined as:

$$\mathbf{L} = -\sum_{i=1}^{N} log(\hat{P}(Y_i|X_i)) \tag{15}$$

where $\hat{P}$ is the predicted distribution of the ground truth $P$ and $Y_i$ is the true label for input $X_i$. NLL belongs to a class of strictly proper scoring rules [28]. A scoring rule is strictly proper if it is uniquely optimized by only the true distribution. NLL is the negative of the logarithm of the probability of the true outcome. If the true class is assigned a probability of 1, NLL will be minimum with value 0.

#### A.3.3 Brier

The Brier score [19] measures accuracy of probabilistic predictions. Across all predicted items $N$ in a set of predictions, the Brier score measures the mean squared difference between the predicted probability assigned to possible outcome for $i \in [1, N]$ and the actual outcome.

$$\mathbf{BS} = (1/N) \sum_{t=1}^{N} \sum_{i=1}^{R} (f_{ti} - o_{ti})^2 \tag{16}$$

Where $R$ is number of possible classes, $N$ is overall number of instances of all classes. $f_{ti}$ is the approximated probability of the forecast $o_{ti}$ in one hot encoding. Brier score can be intuitively decomposed into three components: uncertainty, reliability and resolution [29] and it is also a proper scoring rule.

### Table 9: **ResNet18 ECE on CIFAR10/100 Noise**, averaged over 5 seeds

| | CIFAR10 | | | | | CIFAR100 | | | | |
|---|---|---|---|---|---|---|---|---|---|---|
| Noise-level | 1 | 2 | 3 | 4 | 5 | 1 | 2 | 3 | 4 | 5 |
| Ensemble | 0.051 | 0.075 | 0.076 | 0.118 | 0.184 | **0.059** | **0.076** | **0.078** | **0.107** | 0.149 |
| MCDO | 0.076±0.003 | 0.098±0.004 | 0.102±0.002 | 0.164±0.005 | 0.251±0.008 | 0.114±0.002 | 0.147±0.002 | 0.14±0.006 | 0.192±0.009 | 0.255±0.014 |
| ResNet | 0.102±0.001 | 0.141±0.003 | 0.153±0.007 | 0.209±0.011 | 0.293±0.016 | 0.113±0.004 | 0.149±0.005 | 0.152±0.004 | 0.185±0.005 | 0.237±0.01 |
| Ours (GS) | **0.040±0.002** | **0.055±0.003** | **0.060±0.005** | **0.080±0.01** | **0.106±0.012** | 0.067±0.002 | 0.083±0.002 | 0.089±0.004 | 0.116±0.007 | **0.145±0.013** |

### Table 10: **ResNet18 NLL on CIFAR10/100 Noise**, averaged over 5 seeds

| | CIFAR10 | | | | | CIFAR100 | | | | |
|---|---|---|---|---|---|---|---|---|---|---|
| Noise-level | 1 | 2 | 3 | 4 | 5 | 1 | 2 | 3 | 4 | 5 |
| Ensemble | 0.544 | 0.737 | **0.753** | 1.055 | 1.551 | **1.499** | **1.927** | **1.969** | **2.379** | **2.99** |
| MCDO | 0.667±0.02 | 0.845±0.029 | 0.831±0.013 | 1.262±0.033 | 1.947±0.054 | 1.789±0.011 | 2.225±0.012 | 2.236±0.043 | 2.788±0.072 | 3.585±0.119 |
| ResNet | 0.718±0.01 | 0.979±0.019 | 1.043±0.046 | 1.436±0.081 | 2.052±0.133 | 1.782±0.018 | 2.269±0.023 | 2.326 ± 0.029 | 2.773±0.027 | 3.434±0.032 |
| Ours (GS) | **0.531±0.006** | **0.716±0.012** | 0.785±0.013 | **1.007±0.018** | **1.346±0.02** | 1.786±0.013 | 2.208±0.013 | 2.300±0.014 | 2.676±0.015 | 3.215±0.028 |

## A.4 Calibration in the Face of Differing Levels of Noise

We report additional calibration ECE, NLL and Brier results in the face of different levels of corruption using ResNet18 in Tab. 9, 10 and 11 respectively. CIFAR10 and CIFAR100's validation set was corrupted using a library of common corruptions [1] with 5 levels of severity. In Tab. 9, 10 and 11 we show how differing levels of common corruptions effect the calibration of models. Across all levels of corruption our model consistently had the stronger Brier score in CIFAR100 and much strong ECE and NLL on CIFAR10.

## A.5 Calibration in the Face of Rotation

In Tab. 12b, 12a we rotated CIFAR10 and CIFAR100 validation data set by [0, 350] degrees with 10 degree steps in between, the calibration metrics and accuracy were then averaged. For each model 5 seeds were trained, for MCDO 5 passes were done on each model for inference with a dropout rate of $50\%$ as suggested in the original paper and 5 models were ensembled for Deep Ensemble. $\beta'$ for our models were 4 on CIFAR10 and 10 for CIFAR100.

## A.6 Qualitative Comparison: Extended Discussion

**GSD vs. Single Pass Models** The current state-of-the-art single pass models for inference on OOD data, without training on OOD data, are SNGP [9] and DUQ [8]. The primary disadvantages of these models is: **1) Hyperparameter Combinatorics:** Both DUQ and SNGP require many hyperparameters as shown in Tab. 13. SNGP requires the most hyperparameters out of all the single pass models. The large combinatoric scale, in addition to the fact that these hyperparameters must be tuned via pre-training grid search, make these methods costly as a full training procedure with multiple epochs are required before evaluating calibration. Our model only has *one* hyperparameter that is tuned post-training with 1 epoch on validation set. **2) Extended Training Time:** DUQ requires a centroid embedding update every epoch, while SNGP requires sampling potentially high dimensional embeddings of training points for generating the covariance matrix as well as updates to the bounded spectral norm on each training step, thus increasing training time while our model trains in the same amount of time as the model it is applied to.

**GSD vs. Multi-Pass Models** Bayesian MCDO [7] and Deep Ensemble [14] are considered the current state-of-the-art methods for multi-pass calibration. Bayesian MCDO requires multiple passes with dropout during training and inference in order to achieve stronger calibration. Deep Ensembles

### Table 11: **ResNet18 Brier on CIFAR10/100 Noise**, averaged over 5 seeds

| | CIFAR10 | | | | | CIFAR100 | | | | |
|---|---|---|---|---|---|---|---|---|---|---|
| Noise-level | 1 | 2 | 3 | 4 | 5 | 1 | 2 | 3 | 4 | 5 |
| Ensemble | **0.021** | **0.028** | **0.03** | **0.041** | 0.057 | 0.005 | 0.006 | 0.006 | 0.007 | 0.008 |
| MCDO | 0.024±0.0 | 0.03±0.001 | 0.032±0.0 | 0.046±0.001 | 0.065±0.001 | 0.005±0.0 | 0.006±0.0 | 0.006±0.0 | 0.007±0.0 | 0.009±0.0 |
| ResNet | 0.025±0.0 | 0.034±0.0 | 0.038±0.001 | 0.05±0.002 | 0.068±0.002 | 0.005±0.0 | 0.006±0.0 | 0.007±0.0 | 0.007±0.0 | 0.009±0.0 |
| Ours (GS) | 0.022±0.0 | 0.03±0.0 | 0.034±0.0 | 0.043±0.001 | **0.056±0.001** | **0.003±0.0** | **0.005±0.0** | **0.006±0.0** | **0.007±0.0** | **0.008±0.000** |

| | ECE↓ | NLL↓ | Brier↓ | Accuracy↑ | | ECE↓ | NL↓L | Brier↓ | Accuracy↑ |
|---|---|---|---|---|---|---|---|---|---|
| Ensemble | 0.12±0.047 | **2.973±0.833** | **0.008±0.002** | **0.338** | Ensemble | **0.302** | 2.397 | **0.082** | **0.44** |
| MCDO | 0.254±0.005 | 3.78±0.043 | 0.009±0.0 | 0.282±0.002 | MCDO | 0.373±0.001 | 3.08±0.025 | 0.092±0.0 | 0.401±0.002 |
| ResNet18 | 0.215±0.007 | 3.352±0.036 | 0.009±0.0 | 0.311±0.003 | ResNet18 | 0.42±0.009 | 2.941±0.085 | 0.095±0.001 | 0.427±0.006 |
| Ours | **0.097±0.003** | 3.189±0.019 | **0.008±0.0** | 0.299±0.003 | Ours | 0.323±0.006 | **2.211±0.04** | 0.085±0.001 | 0.422±0.005 |

(a) **ResNet18 on CIFAR100 Rotate** over 5 seeds      (b) **ResNet18 on CIFAR10 Rotate** over 5 seeds

Table 13: Model Requirements

| Model | Loss Function | Hyperparameters | Output | Multi-pass Infer |
|---|---|---|---|---|
| Ensemble | CE | Number of Models | LL | True |
| MCDO | CE | Dropout % | LL | True |
| SNGP | CE | Spectral Norm Bound, GP scale & bias & discount factor & covariance factor & field factor & ridge penalty | GP | False |
| DUQ | Multi-BCE | Gradient penalty, RBF sigma, embedding gamma | RBF | False |
| Ours | CE | $\beta'$, error (default = 0.1) | Decomposed LL | False |

**LL**: Linear Layer. **CE**: Cross-Entropy, **BCE**: Binary Cross-Entropy, **GP**: Gaussian Process, **RBF**: Radial Basis Function

requires $N$ times the number of parameters as the single model it is ensembling where $N$ is the number of models ensembled. The obvious disadvantage to Deep Ensembles is that it requires $N$ times as long to train and run inference as its base model. While no model currently beats Deep Ensemble in accuracy on both clean data and corrupted data, we have shown that our model has stronger calibration in the face of certain levels of severity of corruption Tab. 1 and 2. Bayesian MCDO has shown to have stronger calibration than the same model not trained with dropout, but tends to suffer large accuracy drops as well as not being as strong as other single pass models or Deep Ensemble in calibration, even with many passes. Our model empirically suffers minimal accuracy drops when compared to its backbone and in some conditions led to stronger accuracy on corrupted data (Tab. 1 and 2).

## A.7 Generalizability: Extended Table

**Generalizability** We explored how generalizable our method is by applying it to 12 different models and 4 different datasets in Tab. 14. We report results for both variants of our model: **Grid Searched:** grid search $\beta'$ on the validation set to minimize ECE and **Optimized:** optimize $\beta'$ on the validation set via gradient decent to minimize NLL for 10 epochs, similar to temperature scaling. We can see consistently that our model had stronger calibration across all models and metrics, including models known to be well calibrated like LeNet [22]. All models were tested on CIFAR10C and CIFAR100C datasets offered by [1] where the original CIFAR10 and CIFAR100 were pre-corrupted; these were used for consistent corruption benchmarking across all models. All non-CIFAR datasets were corrupted via rotation from angles [0,350] with 10 step angles in between and the average calibration and accuracy was taken across all degrees of rotation. Our models included: DenseNet [23], LeNet [22] and 6 varying sizes of ResNet, which are described in [24]. The datasets we experimented on CIFAR10 [25], CIFAR100 [25], MNIST [26] and SVHN [27], CIFAR10C [1], CIFAR100C [1].

## A.8 Training Parameters and Dataset License

We train all our models using stochastic gradient descent for 200 epochs and a batch size of 128 on RTX 2080 GPUs. We use a starting learning rate of 0.1 and a weight decay of $5.0e-4$. For ResNet18 experiments, we use a cosine scheduler for learning rate. For Wide ResNet-20-10 experiments, we use a step scheduler which multiplies the learning rate at epoch 60, 120 and 160 by 0.2.

The CIFAR10/100 datasets [25] are released under MIT license. The CIFAR10/100C datasets [1] are released under Creative Commons Public license.

## A.9 Introduction to Temperature Scaling

Temperature scaling is a simple form of Platt scaling [30]. Temperature scaling uses a scalar $T$ to adjust the confidence of the softmax probability in a classification model. Following the notation from

Table 14: **Extended Generalizability Experiments** We benchmark our method against the vanilla models using 12 different backbones and 4 different dOPTasets. **Grid Searched (GS):** $\beta'$ grid searched on validOPTion ECE, **Optimized (OPT):** $\beta'$ optimized via SGD on validation NLL.

| model | dataset | Clean | | | | Corrupt/Rotate | | | |
|---|---|---|---|---|---|---|---|---|---|
| | | accuracy↑ | ECE↓ | NLL↓ | Brier↓ | accuracy↑ | ECE↓ | NLL↓ | Brier↓ |
| LeNet5 | Mnist | 96.16% | 0.01 | 0.132 | 0.006 | 33.95% | 0.43 | 4.533 | 0.104 |
| GSD LeNet5 GS | Mnist | **96.86%** | **0.005** | **0.103** | **0.005** | **35.73%** | 0.42 | 4.405 | 0.101 |
| GSD LeNet5 OPT | Mnist | **96.86%** | 0.012 | 0.106 | **0.005** | **35.73%** | 0.406 | **4.173** | **0.01** |
| DenseNet | SVHN | **41.72%** | 0.051 | 1.71 | 0.072 | 14.31% | 0.301 | 3.844 | 0.107 |
| GSD DenseNet GS | SVHN | 41.7% | **0.027** | **1.62** | **0.069** | **14.41%** | 0.287 | **3.134** | 0.106 |
| GSD DenseNet OPT | SVHN | 41.7% | 0.04 | **1.62** | **0.069** | **14.41%** | 0.277 | 3.25 | **0.105** |
| ResNet34 | CIFAR10 | 95.63% | 0.026 | 0.186 | 0.007 | **81.96%** | 0.164 | 1.114 | 0.039 |
| GSD ResNet34 GS | CIFAR10 | **95.9%** | **0.005** | **0.148** | **0.006** | 76.54% | 0.088 | 0.882 | 0.037 |
| GSD ResNet 34 OPT | CIFAR10 | **95.9%** | 0.011 | 0.162 | 0.007 | 76.54% | 0.054 | 0.813 | 0.035 |
| ResNet50 | CIFAR10 | 95.32% | 0.03 | 0.203 | 0.008 | **76.32%** | 0.17 | 1.23 | 0.039 |
| GSD ResNet50 GS | CIFAR10 | **95.82%** | **0.008** | **0.147** | **0.007** | 76.23% | **0.057** | **0.766** | **0.033** |
| GSD ResNet50 OPT | CIFAR10 | **95.82%** | 0.01 | 0.158 | **0.007** | **76.32%** | 0.115 | 0.928 | 0.038 |
| ResNet101 | CIFAR10 | **95.61%** | 0.028 | 0.197 | **0.007** | 77.59% | 0.154 | 1.118 | 0.037 |
| GSD ResNet101 GS | CIFAR10 | 95.62% | **0.007** | 0.158 | **0.007** | **77.21%** | **0.075** | 0.852 | 0.036 |
| GSD ResNet101 OPT | CIFAR10 | 95.62% | **0.007** | **0.155** | **0.007** | **77.21%** | 0.086 | **0.788** | **0.033** |
| ResNet152 | CIFAR10 | **95.7%** | 0.028 | 0.196 | **0.007** | 75.2% | 0.179 | 1.337 | 0.041 |
| GSD ResNet152 GS | CIFAR10 | 95.63% | **0.007** | **0.151** | **0.007** | **76.58%** | 0.058 | 0.765 | 0.033 |
| GSD Resnnet152 OPT | CIFAR10 | 95.63% | 0.01 | 0.154 | **0.007** | **76.58%** | **0.043** | **0.756** | **0.032** |
| ResNet34 | CIFAR100 | **78.81%** | 0.071 | 0.868 | **0.003** | **51.16%** | 0.19 | 2.387 | **0.007** |
| GSD ResNet34 GS | CIFAR100 | 78.02% | **0.037** | 0.938 | **0.003** | 49.27% | **0.098** | 2.361 | **0.007** |
| GSD ResNet34 OPT | CIFAR100 | 78.02% | 0.043 | **0.93** | **0.003** | 49.27% | 0.112 | 2.372 | **0.007** |
| ResNet50 | CIFAR100 | **79.28%** | 0.0746 | 0.861 | **0.003** | 49.71% | 0.213 | 2.477 | 0.007 |
| GSD ResNet50 GS | CIFAR100 | 78.97% | **0.0326** | 0.879 | **0.003** | **50.12%** | **0.08** | **2.264** | **0.006** |
| GSD ResNet50 OPT | CIFAR100 | 78.97% | 0.041 | **0.856** | **0.003** | **50.12%** | 0.110 | 2.28 | 0.007 |
| ResNet101 | CIFAR100 | **80.17%** | 0.092 | 0.846 | 0.003 | **58.19%** | 0.253 | 2.575 | 0.007 |
| GSD ResNet101 GS | CIFAR100 | **79.82%** | 0.034 | 0.834 | 0.003 | 53.14% | **0.082** | **2.11** | **0.006** |
| GSD ResNet101 OPT | CIFAR100 | **79.82%** | 0.038 | **0.829** | 0.003 | 53.14% | 0.092 | 2.114 | **0.006** |
| ResNet152 | CIFAR100 | **80.71%** | 0.0895 | **0.815** | **0.003** | **54.2%** | 0.233 | 2.45 | 0.007 |
| GSD ResNet152 GS | CIFAR100 | 79.85% | **0.0364** | 0.827 | **0.003** | 53% | **0.078** | **2.12** | **0.006** |
| GSD ResNet152 OPT | CIFAR100 | 79.85% | 0.0397 | 0.821 | **0.003** | 53% | 0.087 | **2.12** | **0.006** |

the main paper, let $l$ denotes the logits. The temperature scalar is applied to all classes as following:

$$P(y|x) = \frac{\exp \frac{1}{T} l_y}{\sum_{j=1}^{c} \exp \frac{1}{T} l_j} = \frac{\exp \left( \|\mathbf{w_y}\|_2 \frac{1}{T} \|\mathbf{x}\|_2 \cos \phi_y \right)}{\sum_{j=1}^{c} \exp \left( \|\mathbf{w_j}\|_2 \frac{1}{T} \|\mathbf{x}\|_2 \cos \phi_j \right)} \quad (17)$$

As described in Fig. 1a, the temperature effectively changes the slope of $\|\mathbf{x}\|_2$ from 1 to $\frac{1}{T}$. The temperature parameter is optimized by minimizing negative log likelihood on a validation set while freezing all the other model parameters [4]. Temperature scaling calibrates a model's confidence on IND data and does not change accuracy. However, it does not provide any mechanism to improve calibration on shifted distribution and is inferior to other uncertainty estimation methods in terms of calibration [5].