# OpenReview forum: "A Geometric Perspective towards Neural Calibration via Sensitivity Decomposition"
_NeurIPS.cc/2021/Conference — NeurIPS 2021 Spotlight_

### Official Review · Reviewer_vKqM · 2021-07-16

**Rating:** 6
**Confidence:** 4

**Summary:**

This paper studies the problem of neural calibration from a novel geometric perspective and finds that the insensitive norm is responsible for bad calibration under distribution shift. Based on this insight, this paper proposes Geometric Sensitivity Decomposition (GSD), which decomposes the norm and angular similarity into an instance-dependent and instance-independent component. Subsequently, this paper proposes a training and inference scheme, to encourage the norm to reflect distribution changes. The proposed method is easy to implement and achieves SOTA results in calibration metrics in the face of corruptions.

**Ethical Concerns:**

Nil

**Limitations And Societal Impact:**

Nil

**Main Review:**

Originality: This work studies the problem of neural calibration under distribution shift from a novel geometric perspective. This work leverages properties of the norm of the input embedding to strengthen calibration, which clearly differs from previous works. However, this work refers to [10][11][12] (references of the paper), the difference between them is not made clear.
Quality: Overall, this work is technically sound and the claims are well supported by experimental results. The finding that the insensitive norm is responsible for bad calibration under distribution shift is key to this work, we hope the authors can give more theoretical analysis or experimental results (e.g., different datasets and base models) to support and strength this point. Meanwhile, some definitions are confusing and should be further explained: why Eq.2 uses “+” while Eq.3 uses “-“? why use such a kind of non-linear function in Eq.9?
Clarity: This paper is well organized. Some writing details need to be worked on:
1.hard-to-understand sentences: line140, “…making the angles larger”; line206, ”…=µx – σx”
2. misquotation:Line255, “Tab9”
3. a table like Tab. 10 may be better to compare “Extended Training Time”.
Significance: Studying the problem of calibration under distribution shift from a geometric perspective has some implications for other researchers or practitioners. This paper achieves promising performance on standard calibration metrics in the face of out-of-distribution (OOD) data and corruption. This problem and method have positive impact. Since I am not very familiar with the neural calibration literature, I am not sure the novelty of this paper reaches the bar of "accept".
The possible limitations and future works of this paper needs to be discussed.

**Time Spent Reviewing:**

24 hours

---

> ### Author Response · Authors · 2021-08-10
> **We provide clarification to notations and provide more experimental support for observations**
>
> We thank the reviewer for the appreciation of the originality of the paper and interesting questions. We would like to provide more experimental evidence to support our assumptions and clarify your questions regarding notations and novelty.  We will also address the writing problems in the revised paper.
>
>
> **1. More support for the observation that insensitive norm is responsible for bad calibration**
>
> Yes, the observation in sec 4.2 holds for other noises and architectures. We further present Pearson correlation between ECE and similarity or norm on 4 models and 3 noises. All models use CIFAR100. A large correlation coefficient indicates a higher positive correlation. Norm is consistently positively correlated with ECE whereas the similarity is either negatively or not correlated with ECE. This shows that the worsening of ECE (large ECE) is correlated with the increasing norm.
>
> Pearson Correlation of Sim/Norm vs. ECE during training
>
> ||||||||||||||
> |----------------	|:------:|:------:|:------:|:------:|:------:|:------:|:------:|:------:|:------:|:------:|:------:|:------:
> | ||ResNet18| ||ResNet-34| ||ResNet101| ||ResNet152||
> | |Shot | Gaussian | Defocus |Shot | Gaussian | Defocus |Shot | Gaussian | Defocus |Shot | Gaussian | Defocus |
> |Sim| 0.09 | 0.03 | 0.73 | 0.09 | 0.03| 0.32| -0.03| -0.04 | -0.88 | -0.97| 0.04| -0.81|
> Norm |**0.82** | **0.82**| **0.78**| **0.82**| **0.81**| **0.78**| **0.87**| **0.87**| **0.85**| **0.86**| **0.85**|**0.81**|
>
> **2. the $+$ and $-$ sign in Equation 2 and 3**
>
> Eq. 2 ($||x|| = ||\Delta x|| + C_x$) is norm decomposition and Eq.3 ($|\phi| = |\Delta \phi| - |C_\phi|$) is angle decomposition. The difference in sign is due to the fact that angular/cosine similarity ($\cos \phi$) is inversely related to angle ($\phi$) in the range 0 to $\pi$. Intuitively, in this range larger $\phi$ results in smaller $\cos \phi$. In other words,  to relax angle means to force $|\Delta \phi|$ to be greater than $|\phi|$, whereas to relax norm means to make $||\Delta x||$ smaller than $||x||$. The absolute value is to take into account angles in the range of 0 to $-\pi$.
>
> **3. Why exponential in equation 9?**
>
> The non-linear function is definitely not unique and is a design choice. The exponential function is used because of its behavior in the limit and better interpretability. Specifically, as the norm of an input increases, the exponential function converges to the original linear function; when the norm is more than one standard deviation below the in-distribution statistics, our model becomes exponentially less confident in prediction; Furthermore, the function goes to zero as the norm goes to zero.
>
> **4. Novelty**
>
> The fact that linear layers can be written as norm and similarity, i.e., $l_i = <x,w_i> = |x||w_i|\cos\phi_i$, is well-known [10,11,12]. However, none of the works studies the effects of norm and similarity on uncertainty estimation and methods to improve their sensitivity. [10] observes that norm models intra-class variation and angle models semantic difference. [11] observes that angle correlates more to the human judgment of sample hardness. [12] analytically studies the influence of angle and norm in the final softmax decision. Our paper is the first to propose sensitivity decomposition of the norm and similarity in an attempt to improve a model’s deterministic uncertainty estimation. The novelty of the approach is two-fold.
>
> First, It opens up **a new direction** to sensitivity modeling in neural networks. Prior works [8,9] focused on Lipschitz-based regularization. We propose a geometric perspective that has never been explored before in this field.
>
> Second, the norm and angle decomposition is **the first attempt** to improve sensitivity under the geometric perspective. We analytically derive a parameterization technique to enforce the decomposition in training with minimum change to existing model architecture and training schemes. Furthermore, it is more interpretable and requires no pre-training hyperparameter tuning compared to prior works [8,9].
>
> [8] Van Amersfoort, Joost, et al. "Uncertainty estimation using a single deep deterministic neural network." International Conference on Machine Learning. PMLR, 2020.
>
> [9] Liu, Jeremiah Zhe, et al. "Simple and principled uncertainty estimation with deterministic deep learning via distance awareness." arXiv preprint arXiv:2006.10108 (2020).
>
> [10] Liu, Weiyang, et al. "Decoupled networks." Proceedings of the IEEE Conference on Computer Vision and Pattern Recognition. 2018.
>
> [11] Chen, Beidi, et al. "Angular visual hardness." International Conference on Machine Learning. PMLR, 2020.
>
> [12] Upchurch, Paul, et al. "Deep feature interpolation for image content changes." Proceedings of the IEEE conference on computer vision and pattern recognition. 2017.
>
> **5. Limitations and Future Work**
>
> We believe the new geometric perspective in deterministic uncertainty modeling can inspire new research in this field. Our next effort is to develop new uncertainty definitions and metrics based on this perspective, which is distinctively different from Bayesian modeling. We also want to apply this method to different learning tasks such as semantic segmentation, automatic speech, and other tasks other than image classification. A limitation of the method is the requirement of post-training tuning of beta on the validation set similar to Platt scaling as well as tuning the error term for the exponential applied to the norm in order to calibrate the model.

---

> > ### Comment · Reviewer_vKqM · 2021-08-22
> > **Response to the author's response**
> >
> > they somewhat solved my concern, and I will keep my decision.

---

### Official Review · Reviewer_3KLr · 2021-07-16

**Rating:** 7
**Confidence:** 3

**Summary:**

The paper proposes Geometric Sensitivity Decomposition (GSD) as the new calibration method in the face of out-of-distribution (OOD) data. The paper identifies that the insensitive norm is responsible for bad calibration under distribution shift. Based on GSD, the paper proposes a simple and quick training and inference scheme. The paper achieves SOTA in calibration metrics in the face of corruptions while having arguably the simplest calibration method to implement.

**Limitations And Societal Impact:**

Good.

**Main Review:**

**[Update after the rebuttal] I am satisfied with the extra mile that the authors went to show the proposed method's effectiveness against the ensemble models. The provided reason behind why it was not a fair comparison before and how it is now fair comparison is sound. Updated my rating accordingly.**

Originality: Are the tasks or methods new? Is the work a novel combination of well-known techniques? (This can be valuable!) Is it clear how this work differs from previous contributions? Is related work adequately cited?

**The method is new. It is clear how this work differs form previous contributions. Related works are adequately cited.**

Quality: Is the submission technically sound? Are claims well supported (e.g., by theoretical analysis or experimental results)? Are the methods used appropriate? Is this a complete piece of work or work in progress? Are the authors careful and honest about evaluating both the strengths and weaknesses of their work?

**Theoretical support is good. Experimental results seem weak given that many metrics are not SOTA compared to prior arts. It is difficult to figure out the pattern when this method yields better results (i.e. Table 3, Table 2); I think it is a work in progress. Yes, honest in reporting all the numbers.**

Clarity: Is the submission clearly written? Is it well organized? (If not, please make constructive suggestions for improving its clarity.) Does it adequately inform the reader? (Note that a superbly written paper provides enough information for an expert reader to reproduce its results.)

**It is clearly written. Well organized.**

Significance: Are the results important? Are others (researchers or practitioners) likely to use the ideas or build on them? Does the submission address a difficult task in a better way than previous work? Does it advance the state of the art in a demonstrable way? Does it provide unique data, unique conclusions about existing data, or a unique theoretical or experimental approach?

**The theoretical idea and its simplicity are promising. The empirical results are not so much as it is presented today.**

**Time Spent Reviewing:**

9

---

> ### Author Response · Authors · 2021-08-10
> **Clarifying the questions on performance**
>
> We thank the reviewer for the appreciation of our theoretical derivation. We would like to provide the following explanation to resolve your questions on the performance of the model.
>
> **1. Performance seems inconsistent compared to Deep Ensemble [2] in table 2 and 3**
> As shown in the paper, our paper consistently outperforms other state-of-the-art deterministic uncertainty estimation (single-pass) methods (DUQ [3], SNGP [1]) in the face of distribution shift. Our model belongs to the class of deterministic uncertainty estimation methods, and therefore a direct comparison to ensemble-based methods is not a fair comparison. Despite that, following prior works in deterministic uncertainty estimation, we included multi-pass methods such as Deep ensemble and MC Dropout as a reference.  For example, Deep ensemble combines the predictions from 10 models. This requires at least 10x the training time, inference time, and parameter storage.
>
> To fairly compete with deep ensemble, we have also included ensembling results of our model, where we trained GSD model on 10 seeds, then tuned beta for all 10 models as an ensemble.  The results for our GSD ensemble model tuning using Grid-Search far outperformed every model and is currently the state-of-the-art method for calibration and accuracy in both distribution shift and clean cifar10 and cifar100. In summary, the single-pass version of our model outperforms previous single-pass methods and the multi-pass version outperforms state-of-the-art Deep Ensemble [2].
>
> ResNet-28-10 on CIFAR10 averaged over 10 seed. $\dagger$ denotes results from [1]. Our method outperforms other single-pass methods and is comparable to Deep Ensemble [2] on corrupted data. While the ensembled version of our model beats all multi-pass models.
>
> |||||||||
> |----------------	|:------:|:------:|:------:|:------:|:------:|:------:|:------:|
> ||Method| Accuracy$\uparrow$| |ECE$\downarrow$| |NLL$\downarrow$|
> |||Clean|Corrupted|Clean|Corrupted|Clean|Corrupted|Clean|Corrupted|
> |Single-Pass| Vanilla$\dagger$  |**96.0**$\pm$0.01 | 72.9$\pm$0.01 |0.023$\pm$0.002 | 0.153$\pm$0.011 | 0.158$\pm$0.01 |1.059$\pm$0.02 | 0.781$\pm$0.01 | 0.835$\pm$0.01 |
> || DUQ$\dagger$ |94.7$\pm$0.02 |71.6$\pm$0.02 |0.034$\pm$0.002 |0.183$\pm$0.011 | 0.239$\pm$0.02 | 1.348$\pm$0.01|
> ||SNGP$\dagger$ |95.9$\pm$0.01 |74.6$\pm$0.01 |0.018$\pm$0.001 |0.090$\pm$0.012 |**0.138**$\pm$0.01 |0.935$\pm$0.01|
> ||Ours $\beta'$ Grid-Searched |95.9$\pm$0.01 | **74.9**$\pm$0.05 | 0.018$\pm$0.003 | **0.067**$\pm$0.01 |0.148$\pm$0.003 | **0.826**$\pm$0.033|
> ||Ours $\beta'$ Optimized |95.9$\pm$0.01 | **74.9**$\pm$0.05 |**0.008**$\pm$0.00 | 0.085$\pm$0.012 | 0.14$\pm$0.004 | 0.853$\pm$0.039|
> |||||||||
> |Multi-Pass| Deep Ensembles$\dagger$ |96.6$\pm$0.01 | **77.9**$\pm$0.01 | 0.010$\pm$0.001 | 0.087$\pm$0.004 | 0.114$\pm$0.01 | 0.815$\pm$0.01|
> ||MC Dropout$\dagger$ |96.0$\pm$0.01 |70.0$\pm$0.02 |0.021$\pm$0,002 | 0.116$\pm$0.009 | 0.173$\pm$0.001 |1.152$\pm$0.01|
> ||Ours $\beta'$Grid-Searched Ensemble | **96.62** |**77.9**| **0.007**| **0.069**| **0.108**| **0.773**|
>
> ResNet-28-10 on CIFAR100 averaged over 10 seed. $\dagger$ denotes results from [1]. Our method outperforms other single-pass methods and is comparable to Deep Ensemble [2] on corrupted data. While the ensembled version of our model beats all multi-pass models.
>
> |||||||||
> |----------------	|:------:|:------:|:------:|:------:|:------:|:------:|:------:|
> ||Method| Accuracy$\uparrow$| |ECE$\downarrow$| |NLL$\downarrow$|
> |||Clean|Corrupted|Clean|Corrupted|Clean|Corrupted|Clean|Corrupted|
> |Single-Pass| Vanilla$\dagger$ | 79.8$\pm$0.02 | **50.5**$\pm$0.04| 0.085$\pm$0.004 | 0.239$\pm$0.020 | 0.872$\pm$0.01 | 2.756$\pm$0.03|
> ||DUQ$\dagger$ |78.5$\pm$0.02 |50.4$\pm$0.02 |0.119$\pm$0.001 |0.281$\pm$0.012| 0.980$\pm$0.02|2.841$\pm$0.01|
> ||SNGP$\dagger$ | **79.9**$\pm$0.03|49.0$\pm$0.02 |0.025$\pm$0.012 |0.117$\pm$0.014 | 0.847$\pm$0.01 | 2.626$\pm$0.01|
> ||Ours $\beta'$ Grid-Searched |79.8$\pm$0.03 | 49.8 $\pm$ 0.003 |**0.027**$\pm$0.003| **0.081** $\pm$ 0.007|  0.787$\pm$0.009 | **2.23**$\pm$0.02|
> ||Ours $\beta'$ Optimized |79.8$\pm$0.03| 49.8$\pm$0.03 | **0.027**$\pm$0.003 | 0.088$\pm$0.007 | **0.784**$\pm$0.011 |2.236$\pm$0.021|
> |||||||||
> |Multi-Pass | Deep Ensembles$\dagger$ | 80.2$\pm$0.01 | **54.1**$\pm$0.04 | 0.021$\pm$0.004 | 0.138$\pm$0.013 | 0.666$\pm$0.02 |2.281$\pm$0.03|
> || MC Dropout$\dagger$ |79.6$\pm$0.02 | 42.6$\pm$0.08 | 0.050$\pm$0.003 | 0.202$\pm$0.010 | 0.825$\pm$0.01 | 2.881$\pm$0.01|
> ||Ours $\beta'$ Grid-Searched Ensemble | **83.09** | **54.1**| **0.018** | **0.086** | **0.614** | **2.042**|
>
> [1] Liu, Jeremiah Zhe, et al. "Simple and principled uncertainty estimation with deterministic deep learning via distance awareness." arXiv preprint arXiv:2006.10108 (2020).
>
> [2] Lakshminarayanan, Balaji, Alexander Pritzel, and Charles Blundell. "Simple and scalable predictive uncertainty estimation using deep ensembles." arXiv preprint arXiv:1612.01474 (2016).
>
> [3] Van Amersfoort, Joost, et al. "Uncertainty estimation using a single deep deterministic neural network." International Conference on Machine Learning. PMLR, 2020.

---

> > ### Comment · Reviewer_3KLr · 2021-08-16
> > **Response to the author's response**
> >
> > I am satisfied with the author's response. In the response, the author shows the proposed method's effectiveness against the ensemble models. The provided reason behind why it was not a fair comparison before and how it is now fair comparison is sound. Updated my rating accordingly.

---

### Official Review · Reviewer_7rY8 · 2021-08-03

**Rating:** 7
**Confidence:** 3

**Summary:**

This paper studies the problem of model calibration in the presence of distribution shifts for vision classification tasks. The authors propose a sensitivity decomposition on the last softmax layer of the model. The resulting calibration approach demonstrates improved performance compared to existing calibration methods under distribution shifts.

**Limitations And Societal Impact:**

See comments above for suggestions.

**Main Review:**

Strengths:
1. The paper takes a geometric perspective to study model calibration. The finding that the insensitivity of the norm accounts for the poor calibration under distribution shifts is interesting.
2. The approximate decomposition is elegant and easy to implement. It does not impose additional computational burden.
3. The proposed calibration approach is simple. It achieves good performance without increasing training and inference complexity significantly.


Major comments:
1. Do you have intuition on why the decomposition should be additive besides its simplicity? What happens if a product term is included in the decomposition?
2. I found the interpretation of $\lVert \Delta x \rVert_2$ confusing. For instance, in the first bullet on the top of page 5, when $\alpha$ is fixed, it is clear that a larger $\beta$ corresponds to a larger $\mathcal{C}_x$, but why a smaller $\lVert \Delta x \rVert_2$? Moreover, in Figure 1c, all OOD data have small values of $\lVert \Delta x \rVert_2$ (this is also implicitly assumed in Eq (9)), why can’t they have large values? If a model outputs moderate values on the training set but large values on the test set, this is also a sign of distribution shifts, right?
3. In the proposed calibration method in Sec 3.4, only $\beta$ is tuned on the validation set. Why don't you also tune $\alpha$?
4. I would like to see standard errors in Table 3-5.
5. In Sec 4.2, the findings are for corruptions with Gaussian noise. Do the findings hold under other types of corruptions?
6. It is unclear to me what the metric AUROC is measuring and how it is related to the OOD detection performance. I would like to see more explanation.

Minor comments:
1. The sentence “While both … calibrated to accuracy” on line 156-158 is unclear to me.
2. On line 161, “\Delta \phi_j” —> “\Delta \phi_y”.


-----
Updates: they answered most of my questions, so I decided to raise my score to 7.


**Time Spent Reviewing:**

6

---

> ### Author Response · Authors · 2021-08-10
> **We provide our answers and additional experiments to the six questions in the review.**
>
> We thank the reviewer for the insightful questions and comments. We provide explanations and new experiments in the following sections.
>
> **1. Do you have intuition on why the decomposition should be additive besides its simplicity? What happens if a product term is included in the decomposition?**
>
> The first intuition stems from the additive interpretation of mean and standard deviation.  We usually think that the majority of samples of a random variable falls into the range of mean $\pm$ standard deviation. The sensitive component acts as the “standard deviation” of each instance and the constant offset is the “mean” of all the instances. The second intuition is simplicity.  With additive decomposition, the original norm/angle depends linearly on the decomposed components while a multiplicative decomposition results in non-linear dependence of the original norm/angle on the decomposed components during training.
> If a product term is included, Eq.6 would have a constant inside the cos term. This will make implementation more difficult as well.  It's easy to extract just the cosine similarity term because the $i$-th logit $l_i = |x||w_i|\cos\phi$. So $\cos\phi$ can be extracted as $l_i/(|x||w_i|)$.  To implement a constant term inside the $\cos$, we would have to do $cos(c*\cos^{-1}(l_i/|x||w_i|))$.
>
> **2. Why larger beta results in smaller $||\Delta x||$? Why OOD data have smaller $||x||$ (or $||\Delta x||$)?**
>
> As shown in Eq.6, $\beta = \frac{1}{\cos C_\phi}C_x$ is proportional to $C_x$ while $C_x$ is inversely linearly related to $||\Delta x||$ in Eq. 2 ( $||x|| = ||\Delta x|| + C_x$).  Because $||x||$ is a constant (for a particular input x), as $C_x$ increases, $||\Delta x||$ decreases. Overall, a larger beta results in smaller $||\Delta x||$.
>
> The assumption that OOD data have smaller norms is based on the expectation that a model should be less confident on OOD data and the construction and training of linear-softmax classifiers. Practically, the norm $||x||$ acts as a temperature in softmax as shown in Eq.1. Intuitively, larger $||x||$ always yields more peaked/confident predictions, and smaller $||x||$ always yields flatter predictive distributions. Therefore, we expect less confident data such as OOD data to have smaller $||x||$ because we expect the output distribution to be flatter. Because $||x||$ is linearly related to $||\Delta x||$ as shown in Eq.2 ( $||x|| = ||\Delta x|| + C_x$), smaller $||x||$ translates to smaller $||\Delta x||$.
> The assumption is supported by the following empirical evidence. In the table below we show the norm of in-distribution and out-of-distribution data on CIFAR10 using ResNet50-GSD (ours). The OOD data is produced by the 15 corruptions used in the paper. OOD data have consistently smaller norms and the accuracy decreases with decreasing norm with a Pearson correlation of 0.9 as an indicator of more out-of-distribution.
>
> |                	| ID	|   | |||OOD |||||||||||||
> |----------------	|:---------------:	|:------: |:------:	|:------:|:------:|:------:|:------:|:------:|:------:|:------:|:------:|:------:|:------:|:------:|:------:|:------:|:------:|:-----------:|
> |  ResNet50-GSD  	|  Clean  | brightness | fog | elastic | snow | defocus| frost |motion| jpeg | zoom| pixelate | contrast | shot| glass |impulse|Gaussian|
> | Norm| **0.73** | 0.66 | 0.52 | 0.42 | 0.46 | 0.46 | 0.44 | 0.37 | 0.39 | 0.35 | 0.5  | 0.39|  0.34| 0.27 | 0.3 | 0.28|
> | Accuracy $\uparrow$| **95.33** | 93.82 | 88.75 | 85.09 | 83.87 | 82.95 | 80.41 | 79.71 | 79.31 | 78.48 | 76.4 | 75.29|59.46|59.29|57.26| 47.33|
>
> Table organized in decreasing accuracy order.
>
> **3. Why do not tune alpha?**
>
> There are 2 reasons for not tuning $\alpha$. First, $\alpha$ is regularized during training so its value is constrained. In our case, it is constrained to 1.  Second, tuning $\alpha$ does not yield additional improvement. We have tried to include $\alpha$ in the tuning process and it does not yield additional improvement.
>
> **4. Standard Deviation for Table 3-5**
>
> We provide standard deviations for tables 3 to 5 calculated from 3 random seeds here. Table 3 has 10 different models on different datasets. We select 3 of them here and will include all std in the revised paper.  After multiple experiments, our conclusions still hold.
>
> Table 3 (Selected): Generalizability Experiments
>
> |                	|||Clean|| |||Corrupt||
> |----------------	|:------: |:------:|:------:|:------:|:------:|:------:|:------:|:------:|:------: |
> | Model | Dataset | Accuracy$\uparrow$ | ECE$\downarrow$ | NLL $\downarrow$ | Brier $\downarrow$ | Accuracy$\uparrow$ | ECE$\downarrow$ | NLL $\downarrow$ | Brier $\downarrow$ |
> |ResNet34| CIFAR10| 95.59 $\pm$ 0.078 | 0.027 $\pm$ 0.001 | 0.188 $\pm$ 0.003  | **0.007** $\pm$ 0.0 | **81.46** $\pm$ 0.576 |  0.173 $\pm$0.009 |1.208 $\pm$ 0.084 | 0.04 $\pm$ 0.02|
> |ResNet34-GSD| CIFAR10| **95.79** $\pm$ 0.09 | **0.011** $\pm$ 0.002 | **0.156** $\pm$ 0.006 | **0.007** $\pm$  0.0 | 79.00 $\pm$ 2.134 | **0.057** $\pm$ 0.011 | **0.801** $\pm$ 0.025 | **0.034** $\pm$ 0.001|
> |ResNet101|CIFAR10| **95.71**  $\pm$ 0.11 | 0.027  $\pm$ 0.001 | 0.191  $\pm$ 0.007 | **0.007**  $\pm$ 0.0 | **80.42**  $\pm$ 2.451 | 0.165  $\pm$ 0.013 | 1.197  $\pm$ 0.11 | 0.039  $\pm$ 0.003|
> |ResNet101-GSD|CIFAR10| 95.58 $\pm$ 0.199 | **0.009** $\pm$ 0.002  | **0.156** $\pm$ 0.006 | **0.007** $\pm$ 0.0 | 79.95 $\pm$ 2.373| **0.061** $\pm$ 0.022 | **0.788** $\pm$0.015 | **0.033** $\pm$ 0.001|
> |ResNet152|CIFAR100| **80.49** $\pm$ 0.43 | 0.083 $\pm$ 0.007 | **0.819** $\pm$ 0.019 | **0.003** $\pm$ 0.000 | **56.37** $\pm$ 1.90 | 0.228 $\pm$ 0.005 | 2.444 $\pm$ 0.012 |  0.007 $\pm$ 0.007|
> |ResNet152-GSD|CIFAR100| 78.82 $\pm$ 0.892 | **0.038** $\pm$ 0.002 | 0.870 $\pm$  0.043 | **0.003** $\pm$ 0.0 | 53.123 $\pm$  2.85 | **0.093** $\pm$  0.008 | **2.217** $\pm$ 0.089 | **0.006** $\pm$ 0.001|
>
> Table4: Importance of Norm
>
> |||||||
> |----------------	|:------:|:------:|:------:|:------:|:------:|
> ||ECE$\downarrow$| NLL$\downarrow$| Brier $\downarrow$| Entropy $\downarrow$| Accuracy$\uparrow$|
> |Vanilla|0.025$\pm$0.001 | 0.186$\pm$0.006 | 0.001$\pm$0.0 | 0.082$\pm$0.002 | 0.954$\pm$0.001|
> |No Weight Norm| 0.061$\pm$0.003 | 0.206$\pm$0.006| 0.001$\pm$0.0 | 0.527$\pm$0.014 | 0.953$\pm$0.001|
> |No x norm | 0.893$\pm$0.002 | 2.837$\pm$0.005 | 0.009$\pm$0.0 | 4.537$\pm$0.001 | 0.954$\pm$0.001|
> |No Cos sim| 0.914$\pm$0.001| 3.235$\pm$0.001 | 0.009$\pm$0.0| 4.546$\pm$0.0| 0.953$\pm$0.001
>
> Table 5: OOD AUROC using Norm and Similarity
>
> |||||
> |----------------	|:------:|:------:|:------:|
> |ResNet18| Criterion | Cifar10  | Cifar10 (Incorrect)|
> |Vanilla| norm | 90.73 $\pm$1.68 | 66.70 $\pm$ 5.33|
> || similarity | 95.65 $\pm$ 2.04 | 60.43 $\pm$ 3.15|
> |$\alpha$-regularized|norm | **98.81** $\pm$ 0.21 | **92.79** $\pm$ 0.82|
> ||similarity | 97.42 $\pm$ 0.30 | 77.97 $\pm$ 2.73|
>
> |||||
> |----------------	|:------:|:------:|:------:|
> |ResNet18| Criterion | Cifar100  | Cifar100 (Incorrect)|
> |Vanilla| norm | 73.82 $\pm$ 6.33 | 53.85 $\pm$ 8.77|
> || similarity | 81.62 $\pm$ 1.64 | 54.36 $\pm$ 2.03|
> |$\alpha$-regularized|norm | **91.92** $\pm$ 3.06 | **81.07** $\pm$ 6.44 |
> ||similarity | 83.21 $\pm$ 2.21 | 59.36 $\pm$ 3.67|
>
> **5. Does the observation in sec 4.2 hold for other noises?**
>
> Yes, the observation in sec 4.2 holds for other noises and architectures. We further present Pearson correlation between ECE and similarity or norm on 4 models and 3 noises. All models use CIFAR100. A large correlation coefficient indicates a higher positive correlation. Norm is consistently positively correlated with ECE whereas the similarity is either negatively or not correlated with ECE. This shows that the worsening of ECE (large ECE) is correlated with the increasing norm.
>
> Pearson Correlation of Sim/Norm vs. ECE during training
>
> ||||||||||||||
> |----------------	|:------:|:------:|:------:|:------:|:------:|:------:|:------:|:------:|:------:|:------:|:------:|:------:
> | ||ResNet18| ||ResNet-34| ||ResNet101| ||ResNet152||
> | |Shot | Gaussian | Defocus |Shot | Gaussian | Defocus |Shot | Gaussian | Defocus |Shot | Gaussian | Defocus |
> |Sim| 0.09 | 0.03 | 0.73 | 0.09 | 0.03| 0.32| -0.03| -0.04 | -0.88 | -0.97| 0.04| -0.81|
> Norm |**0.82** | **0.82**| **0.78**| **0.82**| **0.81**| **0.78**| **0.87**| **0.87**| **0.85**| **0.86**| **0.85**|**0.81**|
>
>
> **6. Why use AUROC for OOD?**
>
>  AUROC is the dominant metric in OOD literature[1,2]. It is used for measuring false positive rate and true positive rate in binary classification by sweeping through a range of thresholds on an uncertainty metric. Specifically, in binary classification using different thresholds will result in different false and true positive rates. AUROC sweeps through a range of thresholds and integrates the area under the true positive vs. false positive curve to generate a scalar summary of a classifier’s performance independent of the threshold. This metric is relevant because OOD detection is a binary classification task. The goal is to classify whether a data is in-distribution or out-of-distribution. In our case, we use norm and angular similarity for the uncertainty metric. Intuitively, a good uncertainty metric will give larger AUROC indicating a better separation between in-distribution and out-of-distribution data.
>
> [1] Mukhoti, Jishnu, et al. "Deterministic neural networks with appropriate inductive biases capture epistemic and aleatoric uncertainty." arXiv preprint arXiv:2102.11582 (2021).
>
> [2] Lakshminarayanan, Balaji, Alexander Pritzel, and Charles Blundell. "Simple and scalable predictive uncertainty estimation using deep ensembles." arXiv preprint arXiv:1612.01474 (2016).

---

### Official Review · Reviewer_ssfF · 2021-08-05

**Rating:** 9
**Confidence:** 4

**Summary:**

The paper presents an approach for calibration of neural networks. The proposed approach is based on the analysis of the softmax logits and decomposing it into two components that depends on the norm of the feature vector (norm component), and the other that depends on the similarity (similarity component) with the weight vector corresponding to a class. The paper then derives learnable parameters that could control each of the components and thus can be used for calibration of the predicted probabilities. The paper claims that for out-of-distribution (OOD) scenarios, this calibration scheme outperforms previously proposed approaches. The empirical results are reported for in-distribution (IND) on CIFAR-10 and CIFAR-100 and their corrupted versions of these for OOD, for various different models.

**Limitations And Societal Impact:**

The limitations and societal impact are pointed to. Perhaps potential failure scenarios of the said method could be pointed to, so that follow up works may explore that as well.


**Main Review:**

Originality:
The idea of geometric decomposition is both interesting and insightful and as far as I know, original. The derivation that leads to trainable parameters that can control the norm and the similarity components is original and seems to be quite generally applicable. This is also validated from the experiments.

Quality:
+ The paper brings new insights over previous approaches [10,11,12] by the proposed geometric decomposition.
+ The experimental results are thoroughly done and presented, validating the claims made.

Clarity:
The paper is very well written and easy to follow. The approach is well motivated and the related work description seems to be complete (AFAIK).

Significance:
I believe the geometric decomposition of the norm component and the similarity component, and its parameterization is an important step towards making predicted probabilities more meaningful. The ideas in this paper could potentially inspire more research as well.

However, I hope the authors can address the following comment in the rebuttal.
- Line 148 states an assumption that \cos|\phi_y| is close to zero. This may be true after sufficient training of a high-capacity model for datasets like CIFAR and other similar ones. However, what happens when this assumption is violated? This may happen for certain classes when there is a high amount of imbalance. Kindly discuss the impact on the performance of the proposed approach.


Update after rebuttal.
My concern has been addressed. I retain the rating. I think it is a pretty good paper.

**Time Spent Reviewing:**

5 hours

---

> ### Author Response · Authors · 2021-08-10
> **We provide additional empirical support for the small-angle assumption and experiments on imbalanced CIFAR10**
>
> We thank the reviewer for the positive comments and appreciation of the proposed method. The question regarding the small-angle assumption is interesting and fundamental in our derivation. We would like to discuss it and provide a way to further enforce its validity. We also conducted an experiment with imbalance cifar10 to specifically answer your questions regarding the effects of imbalance.
>
> **1.Assumption of small-angle in Eq.5**
>
> One of the reasons for the assumption is the observation that high-capacity models tend to be more miscalibrated [1] and our method is especially more effective in this case. When a model is sufficiently high-capacity compared to the diversity of the dataset, the assumption of small-angle is empirically more valid and the method can provide more significant improvement. All the ResNet models are high-capacity deep models and the cosine similarity is close to one as assumed in the paper. Please refer to the following table for empirical support.
>
> The average angle of ground truth class on train data set after training for 200 epochs:
>
> ||||||||
> |----------------	|:------:|:------:|:------:|:------:|:------:|:------:|
> | ||CIFAR10| ||CIFAR100||
> || ResNet-18 | ResNet-34 | ResNet-101 | ResNet-18 | ResNet-34 | ResNet-101 |
> |$\cos \phi$| 0.81 |0.79 | 0.76 | 0.75 | 0.78 |0.74|
>
> **2.Impact of Imbalance**
>
> Imbalance is a very interesting setting. We conduct the following experiment with ResNet18-GSD (Ours) trained on an imbalance cifar10 dataset.  Following standard setting[2], we adopt step imbalance with an imbalance ratio of 100:1. The first 5 classes have 5000 samples each while the last 5 classes only have 50 samples each. In the third and fourth row, we report the average cosine similarity and norm of the training data for each class.  In the last row, we report CIFAR10 vs. SVHN OOD detection results using entropy for each class. The table shows that imbalance does not affect the small-angle assumption but degrades OOD detection performance, especially in the minority classes. The reason is that the small-angle assumption is placed on the training data, the model can fit the training data very well despite imbalance and worse performance during inference.
>
> ||||||||||||
> |----------------	|:------:|:------:|:------:|:------:|:------:|:------:|:------:|:------:|:------:|:------:|
> |ResNet18-GSD(Ours) |5000|5000|5000|5000|5000  |50|50|50|50|50|
> ||airplane | car |bird |cat | deer | dog | frog | horse | ship | truck|
> |$\cos \phi$| 0.79 | 0.85 | 0.77 | 0.78 | 0.81 | 0.80 | 0.81 | 0.80 | 0.83 | 0.81|
> |Norm | 1.08 | 1.63 | 1.08 | 0.97 | 1.39 | 2.48 | 1.26 | 1.73 | 1.98 | 1.80|
> |OOD AUROC $\uparrow$| 97.38 | 99.26 | 96.74 | 94.98 | 98.21 | 90.26 | 80.83 | 82.91 | 89.82 |89.87|
>
>
>
> [1] Guo, Chuan, et al. "On calibration of modern neural networks." International Conference on Machine Learning. PMLR, 2017.
>
> [2] Cao, Kaidi, et al. "Learning imbalanced datasets with label-distribution-aware margin loss." arXiv preprint arXiv:1906.07413 (2019).

---

### Author Response · Authors · 2021-08-10
**Highlights of the Rebuttal**

Thank you to all the reviewers and AC for their hard work during this difficult time. We appreciate the positive comments on the proposed geometric decomposition, and believe this original and insightful new perspective can inspire more research in the area of deterministic uncertainty estimation.  We have thoroughly read through each comment and included new experiments to reflect the suggestions from the reviewers. We highlight some new results here and encourage the reviewers to take a look at the new additions.

1. Our model is a deterministic (single-pass) method and outperforms previous state-of-the-art methods. However, to further demonstrate the power of our approach, we experiment with an ensemble version of our model and it outperforms Deep Ensemble. Please see the response to reviewer **3krl**.

2. We include a correlation study between the ECE and the norm on multiple noises and architectures to further strengthen the observation that insensitive norms are responsible for bad calibration. Please refer to the response to reviewer **7ry8** and **vkqm**.

3. We add empirical supports to the assumption that the cosine similarity between the correct class to its corresponding classifier ($\cos\phi_i$)  is close to one, which is a core assumption in our theoretical derivation. Please refer to the response to reviewer **ssff**.

4. We conduct a new experiment with imbalanced CIFAR10 and discussed the effect of imbalance on our method.  Please refer to the response to reviewer **ssff**.

---

### Decision · Program_Chairs · 2021-09-27

**Decision:**

Accept (Spotlight)

**Comment:**

The authors study the problem of model calibration in the presence of distribution shifts for computer vision. The authors propose a sensitivity decomposition on the last softmax layer of the model. This calibration approach demonstrates improved performance compared to existing ones.

All reviewers recommended to accept.

Accept.